# A versatile system to record cell-cell interactions

Rui Tang[1]*, Christopher W Murray[2], Ian L Linde[3], Nicholas J Kramer[1,4], Zhonglin Lyu[5], Min K Tsai[1], Leo C Chen[1], Hongchen Cai[1], Aaron D Gitler[1], Edgar Engleman[2,3,6], Wonjae Lee[5], Monte M Winslow[1,2,6]*

[1]Department of Genetics, Stanford University School of Medicine, Stanford, United States; [2]Cancer Biology Program, Stanford University School of Medicine, Stanford, United States; [3]Immunology Program, Stanford University School of Medicine, Stanford, United States; [4]Neuroscience Program, Stanford University School of Medicine, Stanford, United States; [5]Department of Neurosurgery, Stanford University School of Medicine, Stanford, United States; [6]Department of Pathology, Stanford University School of Medicine, Stanford, United States

**Abstract** Cell-cell interactions influence all aspects of development, homeostasis, and disease. In cancer, interactions between cancer cells and stromal cells play a major role in nearly every step of carcinogenesis. Thus, the ability to record cell-cell interactions would facilitate mechanistic delineation of the role of the cancer microenvironment. Here, we describe GFP-based Touching Nexus (G-baToN) which relies upon nanobody-directed fluorescent protein transfer to enable sensitive and specific labeling of cells after cell-cell interactions. G-baToN is a generalizable system that enables physical contact-based labeling between various human and mouse cell types, including endothelial cell-pericyte, neuron-astrocyte, and diverse cancer-stromal cell pairs. A suite of orthogonal baToN tools enables reciprocal cell-cell labeling, interaction-dependent cargo transfer, and the identification of higher order cell-cell interactions across a wide range of cell types. The ability to track physically interacting cells with these simple and sensitive systems will greatly accelerate our understanding of the outputs of cell-cell interactions in cancer as well as across many biological processes.

**\*For correspondence:**
tangrui@stanford.edu (RT);
mwinslow@stanford.edu (MMW)

**Competing interests:** The authors declare that no competing interests exist.

## Introduction

Cell-cell interactions contribute to almost all physiological and pathological states (*Deb, 2014*; *Komohara and Takeya, 2017*; *Konry et al., 2016*; *Zhang and Liu, 2019*). Despite the explosion of interest in uncovering and understanding cellular heterogeneity in tissues and across disease states, the extent to which cell-cell interactions influence cell state, drive heterogeneity, and enable proper tissue function remains poorly understood (*Konry et al., 2016*; *Tsioris et al., 2014*; *Zhang and Liu, 2019*). Detailed analysis of the impact of defined cell-cell interactions has illuminated critical aspects of biology; however, these analyses have been limited to a small number of juxtacrine signaling axes that are tractable to study (*Dustin and Choudhuri, 2016*; *Meurette and Mehlen, 2018*; *Yaron and Sprinzak, 2012*).

Interactions between cancer cells and stromal cells play central roles in cancer initiation, progression, and metastasis (*Kitadai, 2010*; *Orimo and Weinberg, 2006*). While secreted factors relaying pro- or anti-tumorigenic signals have been extensively investigated, the impact of direct physical interactions between cancer cells and stromal cells remains understudied (*Bendas and Borsig, 2012*; *Dittmer and Leyh, 2014*; *Nagarsheth et al., 2017*). A greater understanding of the constellation of direct interactions that cancer cells undergo will not only deepen our understanding of tumor ecology but also has the potential to uncover novel therapeutic opportunities (*Nagarsheth et al.,*

**eLife digest** It takes the coordinated effort of more than 40 trillion cells to build and maintain a human body. This intricate process relies on cells being able to communicate across long distances, but also with their immediate neighbors. Interactions between cells in close contact are key in both health and disease, yet tracing these connections efficiently and accurately remains challenging.

The surface of a cell is studded with proteins that interact with the environment, including with the proteins on neighboring cells. Using genetic engineering, it is possible to construct surface proteins that carry a fluorescent tag called green fluorescent protein (or GFP), which could help to track physical interactions between cells.

Here, Tang et al. test this idea by developing a new technology named GFP-based Touching Nexus, or G-baToN for short. Sender cells carry a GFP protein tethered to their surface, while receiver cells present a synthetic element that recognizes that GFP. When the cells touch, the sender passes its GFP to the receiver, and these labelled receiver cells become 'green'.

Using this system, Tang et al. recorded physical contacts between a variety of human and mouse cells. Interactions involving more than two cells could also be detected by using different colors of fluorescent tags. Furthermore, Tang et al. showed that, alongside GFP, G-baToN could pass molecular cargo such as proteins, DNA, and other chemicals to receiver cells.

This new system could help to study interactions among many different cell types. Changes in cell-to-cell contacts are a feature of diverse human diseases, including cancer. Tracking these interactions therefore could unravel new information about how cancer cells interact with their environment.

*2017*; *Swartz et al., 2012*). Furthermore, how diverse cell-cell interactions differentially impact cancer cells at different stages of carcinogenesis and within different organ environments remains largely uncharacterized.

Molecular methods to profile cell state, including in situ approaches within intact tissues, largely fail to uncover the causal relationship between cell-cell interactions and the underlying biology (*Giladi et al., 2020*; *Halpern et al., 2018*). Computational and experimental methods to characterize cell-cell interactions yield additional layers of dimensionality; however, modalities to capture cell-cell interactions are limited (*Boisset et al., 2018*; *Morsut et al., 2016*; *Pasqual et al., 2018*). Much as diverse systems to detect and quantify protein-protein interactions have revolutionized our biochemical understanding of molecular systems, the development of novel systems to detect and quantify cell-cell interactions will accelerate the mapping of the interaction networks of multicellular systems.

Endogenous cell-cell interactions can result in transfer of surface proteins between cells, mainly through either trans-endocytosis or trogocytosis (*Langridge and Struhl, 2017*; *Li et al., 2019*; *Ovcinnikovs et al., 2019*). Thus, we sought to integrate this phenomenon with fluorescent protein tagging to label cells that have undergone direct interactions. We describe a surprisingly robust system (which we term *GFP-based Touching Nexus* or G-baToN) that enables sensitive and specific interaction-dependent labeling of cancer cells and various primary stromal cells, including endothelial cells, T cells and neurons. We extensively characterize this approach and describe several novel applications of this versatile system.

## Results

### G-baToN enables cell-cell interaction-dependent labeling

To create a system in which a fluorescent signal could be transferred between neighboring cells, we adapted a synthetic ligand-receptor system based on the expression of surface GFP (sGFP) on sender cells and a cell surface anti-GFP (αGFP) nanobody on receiver cells (*Fridy et al., 2014*; *Lim et al., 2013*; *Morsut et al., 2016*). Co-culturing sGFP sender cells with αGFP receiver cells led to GFP transfer and labeling of the receiver cells (*Figure 1A,B* and *Figure 1—figure supplement 1A*). Receiver cell labeling required direct cell-cell contact, active membrane dynamics, and pairing between sGFP and its cognate αGFP receptor (*Figure 1C,D* and *Figure 1—figure supplement 1B, C*). Notably, sGFP transfer was accompanied by reduced GFP on the sender cells, downregulation of

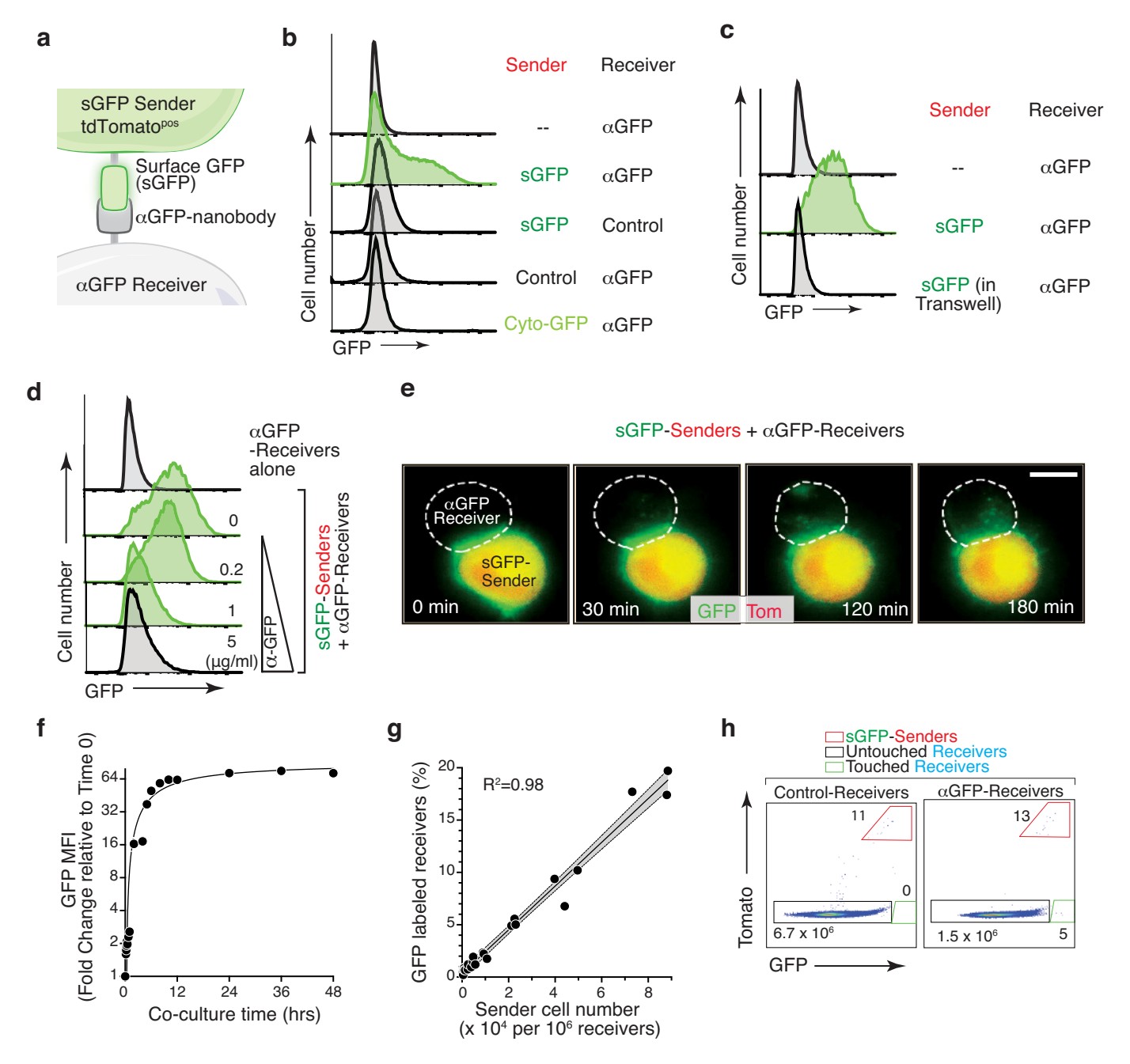

**Figure 1.** GFP-based Touching Nexus (G-baToN) leads to cell-cell interaction-dependent receiver cell labeling. (a) Schematic of the G-baToN system. Surface GFP (sGFP) on a sender cell is transferred to a receiver cell expressing a cell surface anti-GFP nanobody (αGFP) leading to GFP labeling of the 'touched' receiver cell. (b) GFP transfer from sGFP-expressing *KPT* lung cancer sender cells (marked by intracellular tdTomato) to αGFP-expressing 293 receiver cells. Receiver cell labeling is sGFP- and αGFP- dependent. Control sender cells do not express sGFP. Control receiver cells do not express αGFP. Cytoplasmic GFP (Cyto-GFP) is not transferred to receiver cells. Sender and receiver cells were seeded at a 1:1 ratio and co-cultured for 24 hr. Receiver cells were defined as Tomato$^{neg}$PI$^{neg}$ cells. (c) GFP transfer to 293 receiver cells requires direct cell-cell contact. Receiver cells separated from sender cells by a transwell chamber are not labeled. Sender and receiver cells were seeded in upper and lower chambers respectively at a 1:1 ratio and cultured for 24 hr. Receiver cells were defined as Tomato$^{neg}$PI$^{neg}$ cells. (d) GFP transfer to 293 receiver cells requires sGFP-αGFP interaction and is blocked by an anti-GFP antibody in a dose-dependent manner. sGFP sender cells were pre-incubated with the indicated concentration of anti-GFP antibody for 2 hr, washed with PBS, and then co-cultured with receiver cells at a 1:1 ratio for 24 hr. Receiver cells were defined as Tomato$^{neg}$PI$^{neg}$ cells. (e) Time-lapse imaging of GFP transfer from a sGFP-expressing sender cell to an αGFP-expressing receiver cell. Time after contact is indicated. Receiver cell is outlined with white dashed line. Scale bar: 10 μm. (f) Analysis of GFP Mean Fluorescence Intensity (MFI) of αGFP receiver cells (marked by intracellular BFP) co-cultured with sGFP sender cells (marked by intracellular tdTomato) co-cultured for the indicated amount of time. Sender and

*Figure 1 continued on next page*

*Figure 1 continued*

receiver cells were seeded at a 1:1 ratio. Receiver cells were defined as Tomato$^{neg}$PI$^{neg}$BFP$^{pos}$ cells. (**g**) Percentage of labeled αGFP receiver cells after co-culture with different numbers of sender cells for 24 hr. Receiver cells were defined as Tomato$^{neg}$PI$^{neg}$BFP$^{pos}$ cells. (**h**) Detection of rare labeled αGFP receiver cells after co-culture with sGFP sender cells at approximately a 1:10$^{5}$ ratio for 24 hr. Receiver cells were defined as Tomato$^{neg}$PI$^{neg}$BFP$^{pos}$ cells.

The online version of this article includes the following figure supplement(s) for figure 1:

**Figure supplement 1.** GFP transfer requires direct GFP-αGFP interaction.

**Figure supplement 2.** Features of the SynNotch, LIPSTIC, and G-baToN cell-cell interaction reporter systems.

αGFP from the surface of the receiver cells and was partially blocked by chemical inhibitors of endocytosis – all consistent with active GFP transfer and internalization into receiver cells (*Figure 1—figure supplement 1D–F*).

To characterize the kinetics of G-baToN-mediated receiver cell labeling, we performed co-culture time course experiments with time-lapse imaging and flow cytometry readouts. Time-lapse imaging showed rapid transfer and internalization of GFP by receiver cells (*Figure 1E* and *Video 1*). GFP transfer could be detected within five minutes of co-culture and was half-maximal after 6 hr (*Figure 1F* and *Figure 1—figure supplement 1G-H*). Importantly, GFP fluorescence in receiver cells decayed rapidly after isolation of touched receiver cells from sender cells, thus documenting the transient labeling of receiver cells (*Figure 1—figure supplement 1I*). To determine the sensitivity of this system, we co-cultured receiver cells with different ratios of sender cells. The fraction of labeled receiver cells was proportional to the number of sender cells, and even the addition of very few sender cells (representing less than one sender cell to 10$^{5}$ receiver cells) was sufficient to label rare receiver cells (*Figure 1G,H*). Thus, the transfer of GFP to αGFP-expressing cells is a rapid and sensitive method to mark cells that have physically interacted with a predefined sender population.

## Fluorescence transfer efficiency is modulated by transmembrane domains and nanobody affinity

To further characterize the interaction reporter system, we deconstructed the G-baToN design into three functional modules: (1) the transmembrane domain of αGFP on the receiver cells; (2) the pairing between GFP and αGFP; and (3) the transmembrane domain of sGFP on the sender cells. We initially used a published sGFP-αGFP pair in which the Notch1 transmembrane domain links the LaG17-αGFP nanobody onto the receiver cell surface and the PDGFR transmembrane domain links sGFP onto the sender cell surface (*Morsut et al., 2016*). Replacement of the Notch1 transmembrane domain of αGFP with different transmembrane domains allowed us to quantify their impact on GFP transfer efficiency. The VEGFR2 transmembrane domain enabled the highest transfer efficiency, resulting in about a threefold increase relative to the original design (*Figure 2A–C*). We next replaced the LaG17-αGFP nanobody with αGFP nanobodies with varying affinity for GFP (*Figure 2D,E*). While nanobodies exhibiting the highest affinities performed similarly, we noted a minimal affinity required for GFP transfer (*Figure 2F*). Overall, the efficiency of GFP transfer correlated with GFP affinity. Lastly, permutation of the transmembrane domain of sGFP on the sender cell revealed that the rate of retrograde transfer of αGFP-VEGFR2-BFP from receiver to sender cells was influenced by the sGFP transmembrane domain (*Figure 2G–I*). The PDGFR transmembrane domain minimized bidirectional transfer and thus was the optimal design for minimizing retrograde transfer which

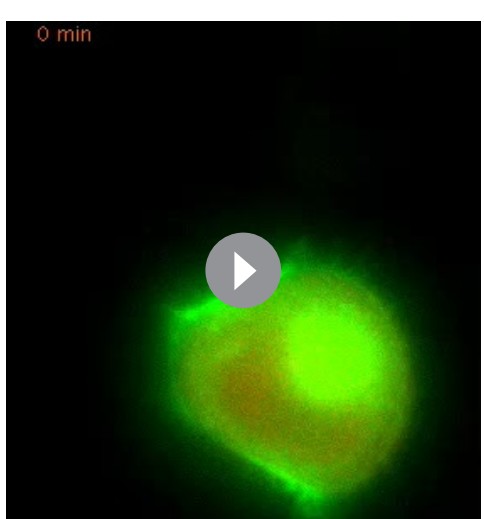

**Video 1.** Time-lapse movie of a sGFP sender cell transferring GFP into a αGFP receiver cell.
https://elifesciences.org/articles/61080#video1

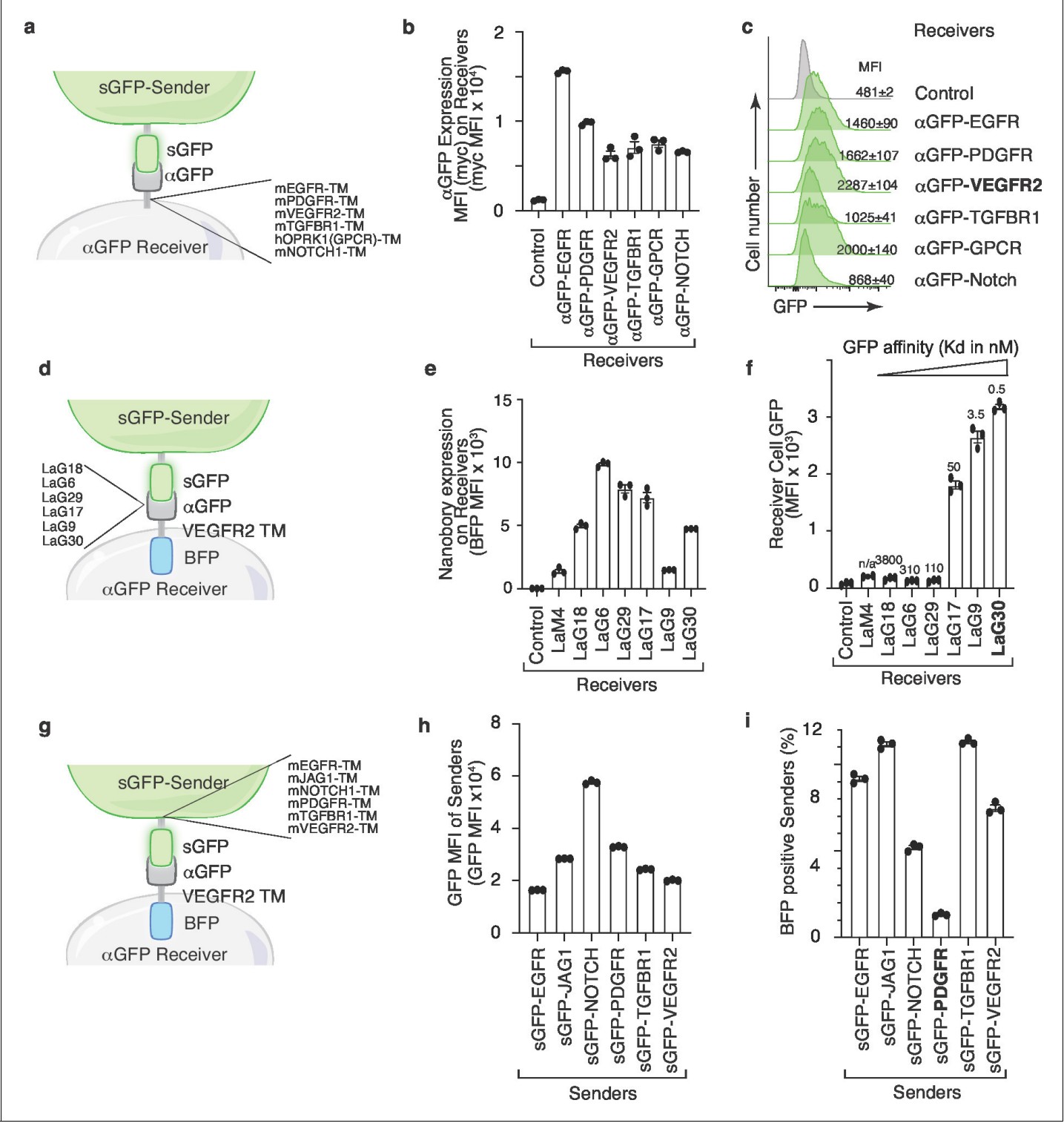

**Figure 2.** Transmembrane domains and the nanobody affinity impact sGFP transfer and receiver cell labeling. (a) Schematic of the sender and receiver cells used to determine the impact of different αGFP transmembrane (TM) domains. TM domains contain the TM domain itself as well as membrane proximal regions from the indicated mouse (m) and human (h) proteins. (b) Different TM domains impact cell surface αGFP expression on 293 receiver cells. Membrane αGFP was quantified by anti-Myc staining. Control receiver cells do not express any nanobody. Mean +/- SD of Myc MFI of triplicate cultures is shown. (c) VEGFR2 TM domain on αGFP receiver cells enable highest GFP transfer efficiency. Receiver cells expressing αGFP linked to different TM domains were co-cultured with sGFP sender cells at a 1:1 ratio for 6 hr. Receiver cells were defined as Tomato[neg]PI[neg] cells. (d) Schematic of the sender and receiver cells used to determine the impact of different αGFP nanobodies on G-baToN-based labeling. (e) Different nanobodies

*Figure 2 continued on next page*

*Figure 2 continued*

exhibit different levels of expression on 293 receiver cells. Total αGFP expression was assessed by BFP intensity. Mean +/- SD of GFP MFI of triplicate cultures is shown. (f) αGFP affinity influences transfer of GFP to touched 293 receiver cells. Receiver cells expressing different αGFP nanobodies were co-cultured with sGFP sender cells at a 1:1 ratio for 6 hr. GFP transfer was assessed by flow cytometry. GFP intensity on Tomato[neg]PI[neg]BFP[pos] receiver cells is shown as mean +/- SD of triplicate cultures. (g) Schematic of the sender and receiver cells used to determine the impact of different sGFP TM domains on G-baToN-based labeling. TM domains contain the TM domain itself as well as membrane proximal regions from the indicated mouse (m) proteins. (h) Different TM domains on sGFP impact its expression in 293 sender cells. sGFP expression in sender cells was assessed by flow cytometry for GFP. Mean +/- SD of GFP MFI of triplicate cultures is shown. (i) PDGFR TM domain on sGFP minimized retrograde transfer of αGFP from receiver cells to 293 sGFP sender cells. αGFP transfer to sGFP sender cells was determined as the percentage of mCherry[pos]GFP[pos] sender cells that were also BFP[pos]. Cells were co-cultured for 6 hr at a 1:1 ratio. Mean +/- SD of triplicate cultures is shown.

could generate false-positive signals (*Figure 2G–I*). Collectively, the permutation of the transmembrane domains anchoring sGFP and αGFP, as well as varying the αGFP nanobody affinity identified designs that maximized unidirectional receiver cell labeling.

## Tracking cancer-stroma interactions using G-baToN

Cancer cells interact with a variety of stromal cells at both the primary and metastatic sites (*Kota et al., 2017*; *Nielsen et al., 2016*). Thus, we employed the G-baToN system to record various cancer-stroma interactions in conventional 2D and 3D microfluidic culture systems as well as in vivo. Co-culturing sGFP-expressing lung adenocarcinoma cells with primary human umbilical vein endothelial cells (HUVECs) in a 2D format led to robust endothelial cell labeling (*Figure 3A,B*). Additionally, within 3D microfluidic chips, pre-seeded HUVECs expressing αGFP were robustly labeled following co-incubation with sGFP-expressing lung adenocarcinoma cells (*Figure 3E–G*). Thus, the G-baToN system is able to efficiently record cancer cell-endothelial cell interactions across multiple culture conditions.

Given the importance of interactions with adaptive immune cells during carcinogenesis (*Crespo et al., 2013*; *Joyce and Fearon, 2015*), we assessed the ability of the G-baToN system to track the interaction of primary human CD4 and CD8 T cells with lung cancer cells. αGFP-expressing CD4 and CD8 T cells that interacted with sGFP-expressing lung cancer cells in culture were specifically labeled (*Figure 4A–C*). To test the ability of the G-baToN system to capture cancer cell-T cell interactions in vivo, we established lung tumors from a sGFP-expressing lung adenocarcinoma cell line prior to intravenous transplantation of αGFP-expressing CD4 T cells. 24 hr after T cell transplantation, over 60% of αGFP-expressing CD4 T cells within the tumor-bearing lungs were labeled with GFP, while control CD4 T cells remained unlabeled (*Figure 4D,E*). Thus, the G-baToN system is capable of recording cancer cell-T cell interactions both in vitro and in vivo.

Recent studies have demonstrated a supportive role for neurons within the primary and metastatic niche in the context of brain (*Venkatesh et al., 2019*; *Zeng et al., 2019*). To determine whether G-baToN can record cancer cell-neuron interactions, we co-cultured sGFP-expressing lung adenocarcinoma cells with primary cortical neurons expressing αGFP. Physical contact between cancer cells and neuronal axons led to punctate-like GFP granule transport into receiver neurons (*Figure 5A–B*). These results demonstrate the successful application of G-baToN system to record a variety of cancer cell-stromal cell interactions.

## G-baToN can be applied in a wide range of cell types

To assess the generalizability of the G-baToN system across cell types, we expressed αGFP in a panel of cell lines and primary cells. Each receiver cell type was able to uptake GFP from sGFP-expressing lung cancer sender cells upon cell-cell contact (*Figure 5—figure supplement 1A*). Furthermore, diverse cancer cell lines and primary cell types expressing sGFP were able to transfer GFP to αGFP-expressing HEK293 receiver cells (*Figure 5—figure supplement 1B–F*). As anticipated, receiver cell labeling required sGFP-expression on the sender cell and αGFP expression on the receiver cells. Thus, G-baToN-based labeling extends beyond transformed cell types and can label diverse primary cell types in co-culture.

To further test the generalizability of the system and determine whether primary cells can serve as both sender and receiver cells, we assessed GFP transfer between interacting primary cells in the context of two well-established heterotypic cell-cell interactions: endothelial cells interacting with

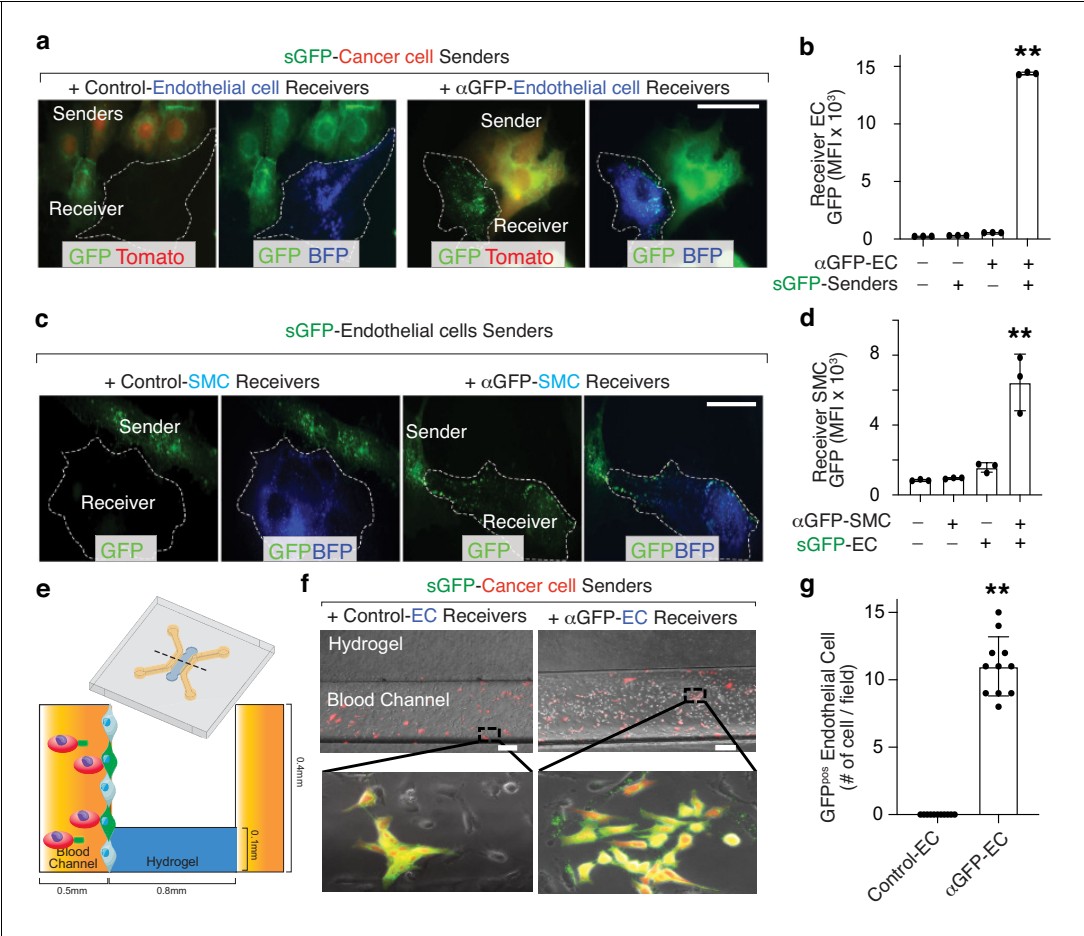

**Figure 3.** G-baToN can be detect cancer cell-endothelial cell and endothelial cell-smooth muscle cell interactions. (**a, b**) G-baToN can detect cancer cell-endothelial cell (EC) interactions. HUVECs expressing αGFP were co-cultured with or without Tomato^pos sGFP-expressing lung cancer sender cells at a 1:1 ratio for 24 hr. (**a**) Representative images of Tomato^pos sGFP-expressing lung cancer sender cells co-cultured with either control HUVEC receiver cells (HUVECs expressing BFP) or αGFP HUVEC receiver cells at a 1:1 ratio for 24 hr. Scale bars = 50 μm. (**b**) MFI of GFP on PI^neg Tomato^neg BFP^pos CD31^pos Receiver cells was assessed by flow cytometry and is shown as mean +/- SD of triplicate cultures. **p<0.01, n = 3. (**c,d**) G-baToN can detect endothelial cell (EC)-smooth muscle cell (SMC) interactions. Primary human umbilical artery smooth muscle cells (HUASMC) expressing αGFP were co-cultured with or without sGFP-expressing HUVEC sender cells at a 1:1 ratio for 24 hr. (**c**) Representative images of sGFP-expressing HUVEC sender cells co-cultured with either control HUASMC receiver cells (expressing BFP) or αGFP HUASMC receiver cells at a 1:1 ratio for 24 hr. Scale bars = 50 μm. (**d**) MFI of GFP on PI^neg BFP^pos receiver cells was assessed by flow cytometry and is shown as mean +/- SD of triplicate cultures. **p<0.01, n = 3. (**e,f,g**) G-baToN can detect cancer cell-endothelial cell interactions in 3D-microfluidic culture. (**e**) Details on design of 3D-microfluidic devices for cancer cell-endothelial cell co-culture. (**f**) Representative images of Tomato^pos sGFP-expressing lung cancer sender cells co-cultured with either control HUVEC receiver cells (HUVECs expressing BFP) or αGFP HUVEC receiver cells at a 1:10 ratio for 24 hr. Scale bars = 200 μm. (**g**) Average number of GFP^pos HUVEC after co-culture with cancer cells for 24 hr. 10 areas from three chips with 200X magnification were used for the quantification. **p<0.01, n = 10.

smooth muscle cells and astrocytes interacting with neurons. Co-culturing sGFP-expressing HUVEC and αGFP-expressing primary human umbilical vein smooth muscle cells (HUVSMC) resulted in efficient receiver smooth muscle cell labeling (*Figure 3C,D*). Furthermore, sGFP-expressing astrocytes were able to transfer GFP to αGFP-expressing cortical neurons (*Figure 5C,D*). Collectively, these results document the efficiency of G-baToN-based cell labeling across diverse cell types.

## Multicolor labeling enables recording of reciprocal and higher-order interactions

Given the high efficiency with which sGFP labels receiver cells upon interaction with cognate sender cells, we tested whether other surface antigen/antibody pairs could lead to protein transfer and labeling. Due to the cross reactivity of αGFP with BFP, co-culture of surface BFP (sBFP) sender cells

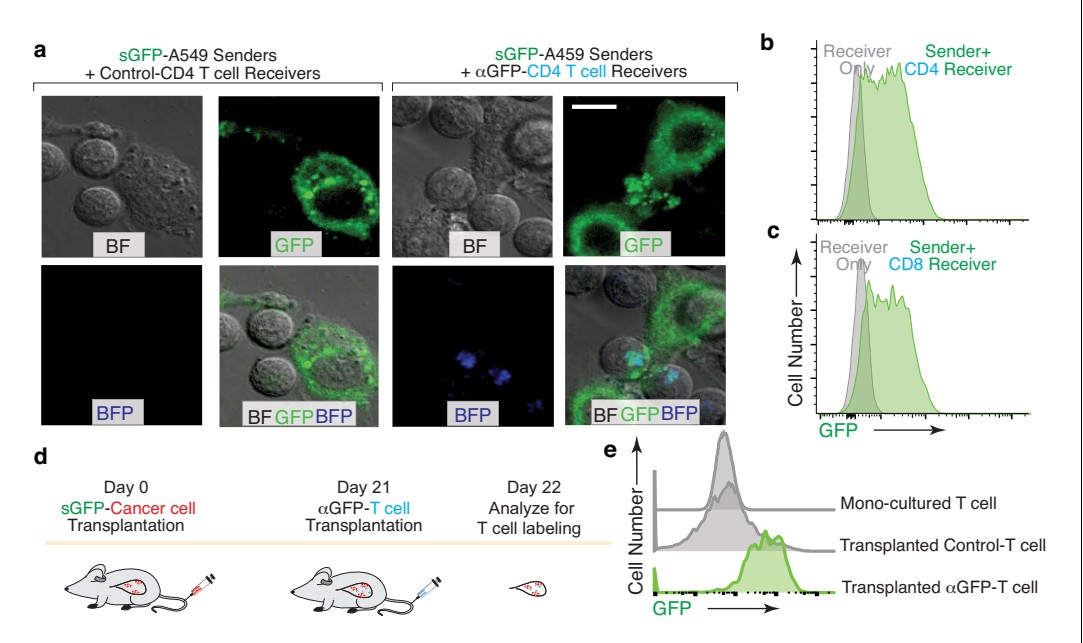

**Figure 4.** G-baToN can detect cancer cells – T cells interactions. (**a,b,c**) G-baToN can detect cancer cell-T cell interactions in vitro. (**a**) Primary human CD4$^{pos}$ T cells were co-cultured with sGFP-expressing lung cancer sender cells (A549 cells) at a 2:1 ratio for 24 hr. Representative image of A549 cell and CD4 T cell interactions. Scale bars = 10 μm. BF = bright field (**b,c**) A549 cells expressing sGFP can transfer GFP to αGFP primary human CD4$^{pos}$ (**b**) or CD8$^{pos}$ (**c**) T cells after co-culture at a 1:1 ratio for 24 hr. Receiver cells were defined as Near-IR$^{neg}$BFP$^{pos}$CD4$^{pos}$ or CD8$^{pos}$ T cells. (**d,e**) G-baToN can detect cancer cell-T cell interactions in vivo. (**d**) Experiment design for cancer cell-T cell interactions in vivo. 1 × 10$^6$ sGFP-expressing lung cancer sender cells were transplanted into NSG mice at day 0. 4 × 10$^6$ αGFP primary human CD4$^{pos}$ T cell were transplanted into tumor-bearing mice at day 21. One day after T cell transplantation (day 22), T cells in the mouse lung were analyzed by FACS. (**e**) sGFP-expressing cancer cell can transfer GFP to αGFP-expressing primary human CD4$^{pos}$ T cells. Receive cells were defined as PI$^{neg}$BFP$^{pos}$CD4$^{pos}$T cells.

with αGFP receiver cells generated BFP-labeled receiver cells (*Fridy et al., 2014*; *Figure 6—figure supplement 1A, B*). Orthogonal systems consisting of surface-mCherry/αmCherry (LaM4) (*Fridy et al., 2014*) and surface-GCN4-GFP/αGCN4 (single-chain variable fragment, scFV) (*Tanenbaum et al., 2014*) also led to efficient and specific receiver cell labeling (*Figure 6—figure supplement 1C–F*). Thus, the G-baToN labeling system can be extended to additional antigen/antibody pairs.

We next integrated these orthogonal systems to enable reciprocal labeling and detection of higher order multi-cellular interactions. Engineering cells with these orthogonal systems in an anti-parallel fashion should enable reciprocal labeling of both interacting cells. Co-culture of cells expressing sGFP and αmCherry with cells expressing smCherry and αGFP resulted in reciprocal labeling of both interacting cell types (*Figure 6A,B*, and *Figure 6—figure supplement 2A*). This reciprocal labeling system may be particularly useful when the interaction elicits changes in both interacting cell types. Using orthogonal ligand-receptor pairs, we also created an AND gate dual labeling strategy. Specifically, co-expression of αmCherry and αGFP on receiver cells enabled dual color labeling of receiver cells that had interacted with smCherry-expressing, sGFP-expressing, or both sender cell types (*Figure 6C,D*, and *Figure 6—figure supplement 2B*). Analogously, we achieved dual-color labeling of receiver cells by leveraging the ability of αGFP to bind to both sGFP and sBFP (*Figure 6E,F*). Thus, derivatives of the G-baToN system allow for additional degrees of resolution of complex cell-cell interactions.

## Labeling with HaloTag-conjugated fluorophores enhances sensitivity and signal persistence

We next extended our labeling system further by generating sender cells expressing the HaloTag protein fused to sGFP (sHalo-GFP; *Figure 7A*; *Los et al., 2008*). Covalent attachment of a synthetic fluorophore to sHalo-GFP enabled specific loading onto sender cells (*Figure 7B*). Co-culture of

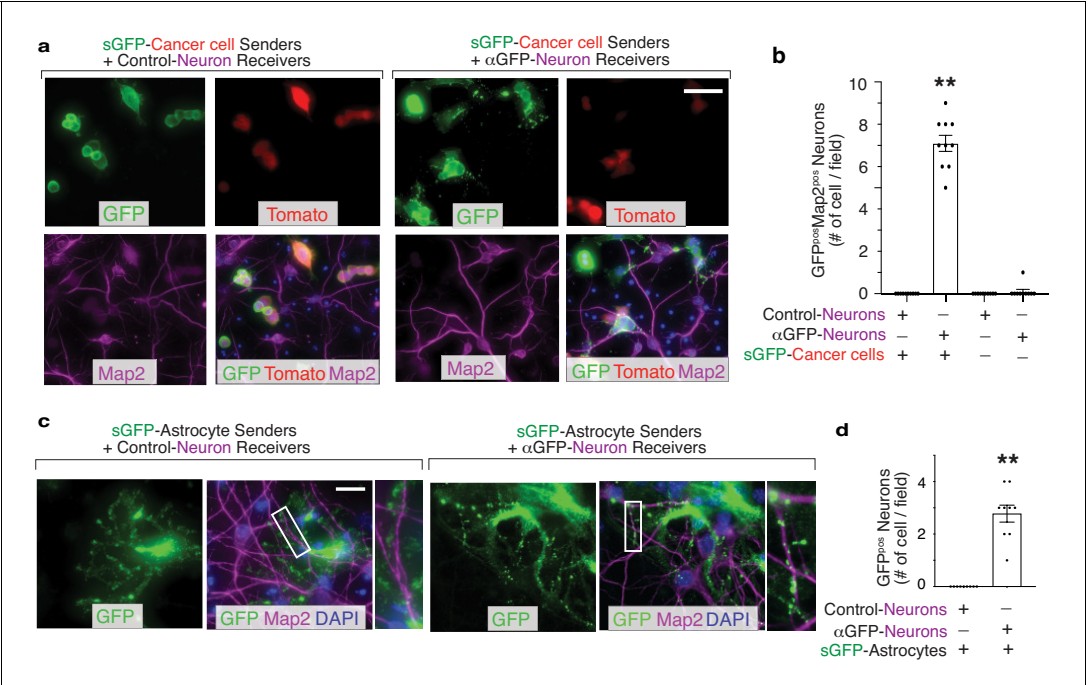

**Figure 5.** G-baToN can detect cancer cell–neuron and astrocyte-neuron interactions. (**a**) Representative image of sGFP-expressing cancer sender cells co-cultured with either control neuron receiver cells or αGFP neuron receiver cells at a 1:1 ratio for 24 hr. Neurons were stained with Microtubule Associated Protein 2 (Map2). Scale bars = 50 µm. (**b**) Quantification of a using images from 10 different fields. Each dot represents a field. The bar indicates the mean +/- SD. GFP$^{pos}$ neurons were defined as Map2$^{pos}$Tomato$^{neg}$ cells with GFP. **p<0.01, n = 10. (**c**) Representative images of sGFP-expressing astrocyte sender cells co-cultured with either control neuron receiver cells or αGFP neuron receiver cells at a 1:2 ratio for 24 hr. Neurons were stained with Map2. Scale bars = 50 µm. Higher magnification of the boxed areas are shown on the right. (**d**) Quantification of c using images from 10 different fields. Each dot represents a field. The bar indicates the mean +/- SD. GFP$^{pos}$ neurons were defined as Map2$^{pos}$ cells with GFP. **p<0.01, n = 10.

The online version of this article includes the following figure supplement(s) for figure 5:

**Figure supplement 1.** G-baToN is a generalized system that can be used for touching-based labeling between various cell types.

Alexa Fluor 660 (AF660)-loaded sHalo-GFP sender cells with αGFP receiver cells enabled co-transfer of both GFP and AF660 (*Figure 7C*). Compared to GFP, transfer of the chemical fluorophore using sHalo-GFP-based labeling of receiver cells led to increased signal-to-noise ratio and higher sensitivity (*Figure 7C,D*). Importantly, changing from a protein (GFP) to a chemical fluorophore also extended the half-life of labeling, thus enabling partially tunable persistence of labeling after touching (*Figure 7E*).

Next, we coupled the enhanced properties of chemical fluorophore-based labeling with the generalizability of the GCN4-baToN system to assemble a robust and versatile system to label receiver cells that have interacted with two or more different sender cell types (*Figure 7F*). Co-culturing αGCN4 receiver cells with AF488- and AF660-loaded sGCN4-Halo sender cells generated a spectrum of receiver cells with varying degrees of AF488 and AF660 labeling (*Figure 7G*). Importantly, the ratio of AF488 to AF660 transferred to the dually labeled receiver cells strongly correlated with the ratio of the two sGCN4-Halo sender populations within the co-culture, suggesting that this system can quantitively measure higher order cell-cell interactions (*Figure 7H*).

## The G-baToN system can function as a vehicle for molecular cargo

Given the high efficiency of protein transfer using the G-baToN system, we investigated whether cargo molecules could be co-transferred with GFP from sender cells to receiver cells. In addition to the co-transfer of Halo-Tag with sGFP, we also generated sender cells with surface expression of a GFP-tdTomato fusion protein (sGFP-Tom) and uncovered stoichiometric tdTomato and GFP transfer to αGFP receiver cells (*Figure 8A,B*). Beyond fluorescent labels, we tested whether other cargo

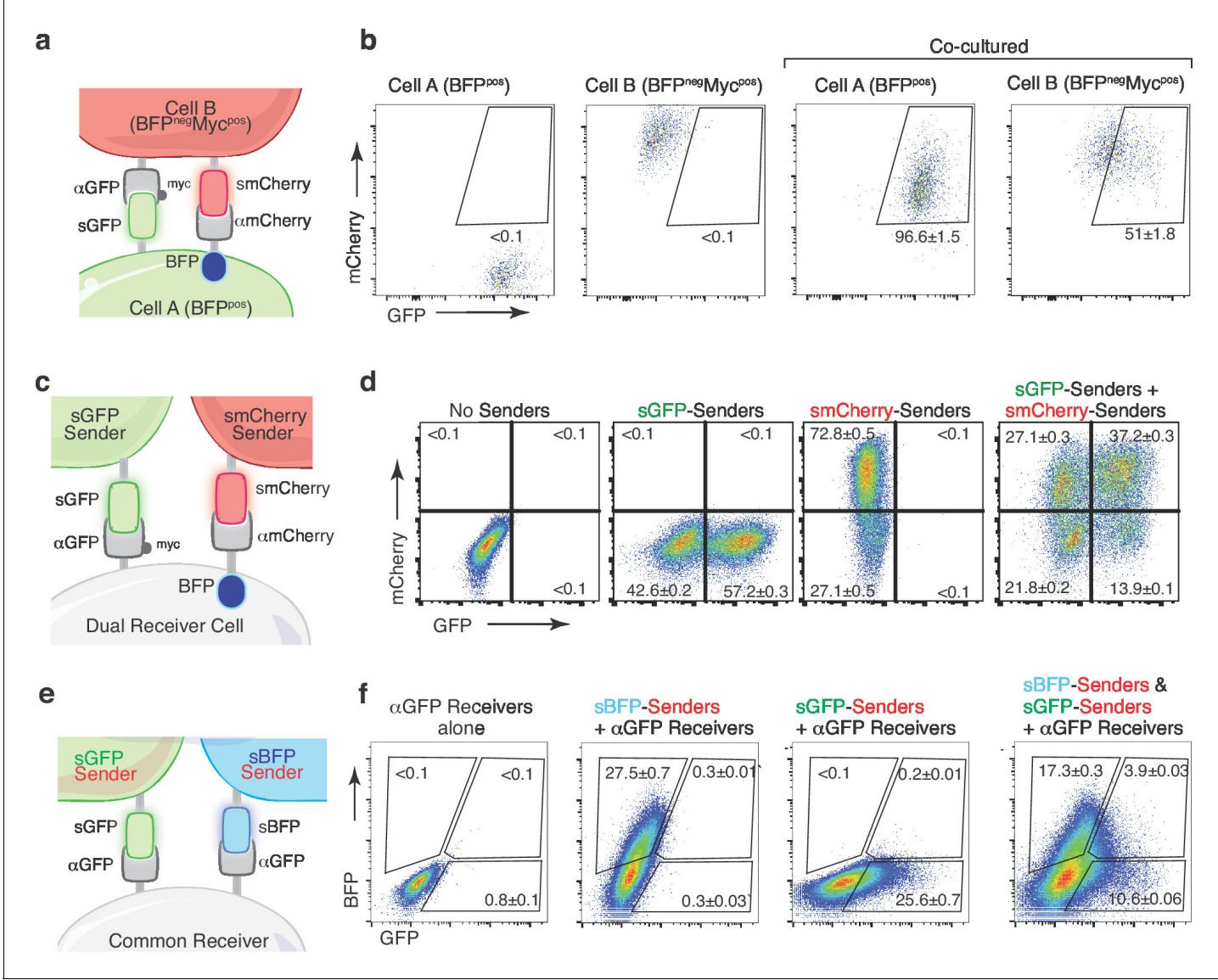

**Figure 6.** Multicolor-baToN systems enable recording of higher-order interactions. (a) Diagram of the reciprocal baToN system. Cell A expresses sGFP and αmCherry (tagged by intracellular BFP), Cell B expresses smCherry and αGFP (tagged by Myc-tag). (b) Representative FACS plots of cell A and cell B monocultures (left two panels) and after co-culture at a 5:1 ratio for 24 hr. Percent of labeled cells is indicated as mean +/- SD of triplicate cultures. (c) Schematic of the AND gate-baToN system. sGFP and smCherry sender cells express either sGFP or mCherry. Dual receiver cells express both αGFP (LaG17, tagged by Myc-tag) and αmCherry (LaM4, tagged by intracellular BFP). (d) Representative FACS plots of dual receiver 293 cells cultured with the indicated 293 sender cells at 1:1 (for single sender cell) or 1:1:1 (for dual sender cells) ratios. Percent of labeled receiver cells (defined as BFP$^{pos}$) after 24 hr of co-culture is indicated as mean +/- SD of triplicate cultures. (e) Diagram of the BFP/GFP AND gate baToN system. sBFP sender cells express intracellular Tomato and surface BFP, sGFP sender cells express intracellular Tomato and surface GFP. Common receiver cells expressed αGFP. (f) Representative FACS plots of common receiver 293 cells cultured with the indicated Tomato$^{pos}$ sender cells at 1:1 (for single sender cell) or 1:1:1 (for dual sender cells) ratios. Receiver cells were defined as Tomato$^{neg}$PI$^{neg}$. Percent of labeled common receiver cells after 24 hr of co-culture is indicated as mean +/- SD of triplicate cultures.

The online version of this article includes the following figure supplement(s) for figure 6:

**Figure supplement 1.** X-baToN systems enable fluorescent labeling via various antigen-nanobody/scFV pairs.

**Figure supplement 2.** Dual color-baToN systems enable labeling in complex cell-cell interaction systems.

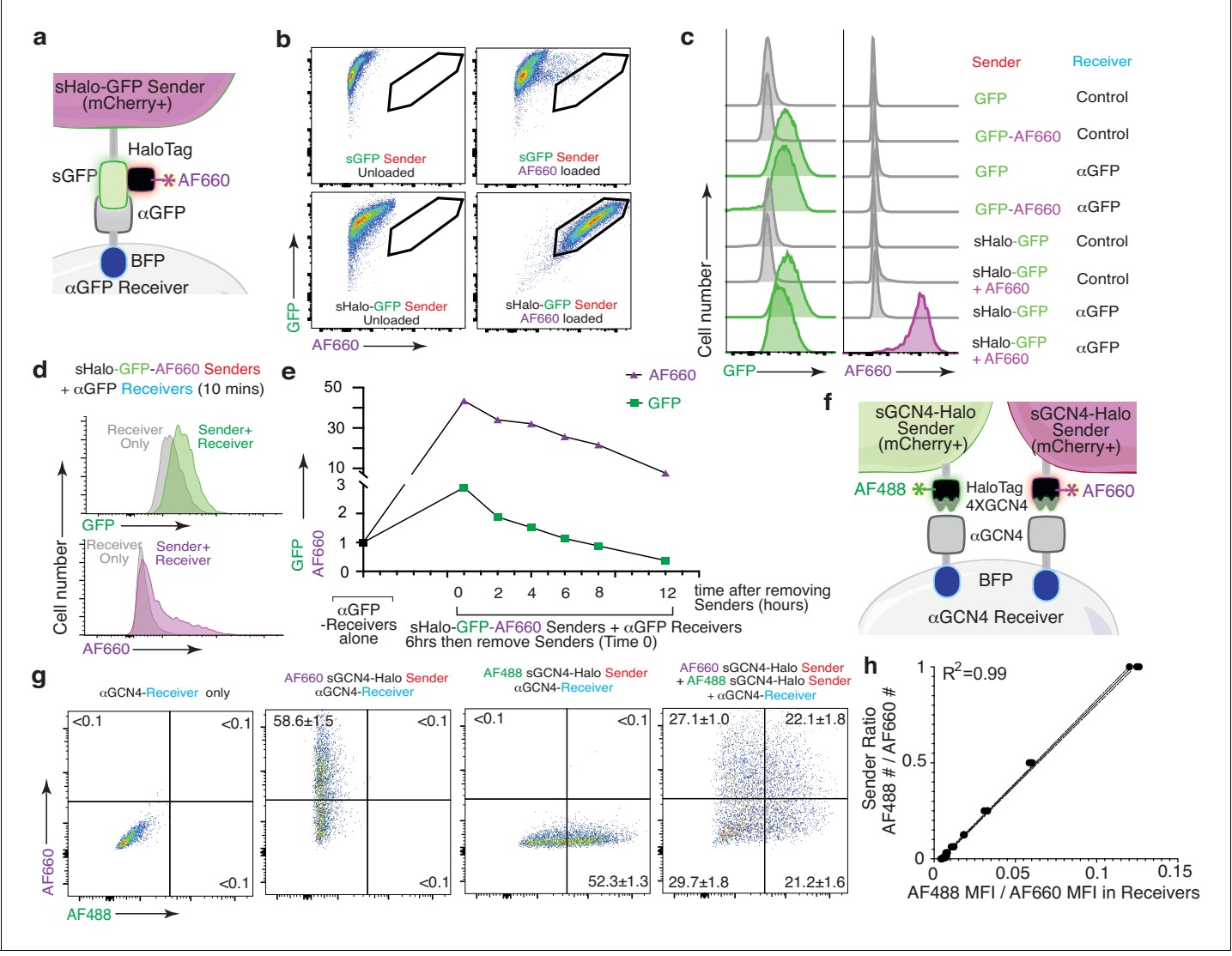

**Figure 7.** The HaloTag-baToN system enables quantitative and sensitive cell-cell interaction-dependent receiver cell labeling. (**a**) Diagram of HaloTag-baToN system. Sender cells (marked by intracellular 2A-mCherry) express surface HaloTag-GFP fusion which can be loaded with HaloTag ligands (in this example AF660). Receiver cells express αGFP (LaG17, tagged by intracellular BFP). (**b**) Labeling of HaloTag-expressing sender cells with AF660 fluorophore. Representative FACS plots of *KP* (lung adenocarcinoma) sender cells expressing either sGFP or sGFP-sHaloTag incubated with AF660-conjugated HaloTag ligand for 5 min on ice. AF660 specifically labeled sHaloTag-GFP sender cells but not sGFP sender cells. (**c**) Representative plot of GFP and AF660 intensity in αGFP 293 receiver cells co-cultured with HaloTag-GFP *KP* sender cells at a 1:1 ratio for 6 hr. Receiver cells were defined as mCherry$^{neg}$PI$^{neg}$BFP$^{pos}$ cells. (**d**) AF660 transfer to αGFP 293 receiver cells is rapid after cell-cell interaction. AF660 MFI shift was detected after mixing sHalo-GFP sender cells and αGFP receiver cells and co-culture for 10 min. AF660 MFI shift was more dramatic than GFP. Receiver cells were defined as mCherry$^{neg}$PI$^{neg}$BFP$^{pos}$ cells. (**e**) Slower AF660 quenching in touched receiver cells after removing sHalo-GFP sender cells. After 6 hr co-culture, GFP/AF660-positive receiver cells were purified via FACS. Analysis of GFP/AF660 MFI in purified receiver cells showed rapid GFP degradation but slower AF660 quenching. Receiver cells were defined as mCherry$^{neg}$PI$^{neg}$BFP$^{pos}$ cells. (**f**) Diagram of dual color GCN4-HaloTag-baToN system. Sender cells (marked by intracellular 2A-mCherry) express surface 4XGCN4 associated with HaloTag, loaded with either AF488- or AF660- conjugated HaloTag ligand. Receiver cells express αGCN4 (tagged by intracellular BFP). (**g**) Representative FACS plots of αGCN4 receiver cells co-cultured with the indicated sender cells at 1:1 (for single sender cell) or 1:1:1 (for dual sender cells) ratios. Percent of labeled receiver cells (gated as mCherry$^{neg}$PI$^{neg}$BFP$^{pos}$) after 6 hr of co-culture is indicated as mean +/- SD of triplicate cultures. (**h**) AF488/AF660 GCN4-HaloTag sender ratio in the co-culture directly proportional to AF488/AF660 intensity (MFI) of αGCN4 receiver after 6 hr of co-culture. Receiver cells were defined as mCherry$^{neg}$PI$^{neg}$BFP$^{pos}$ cells.

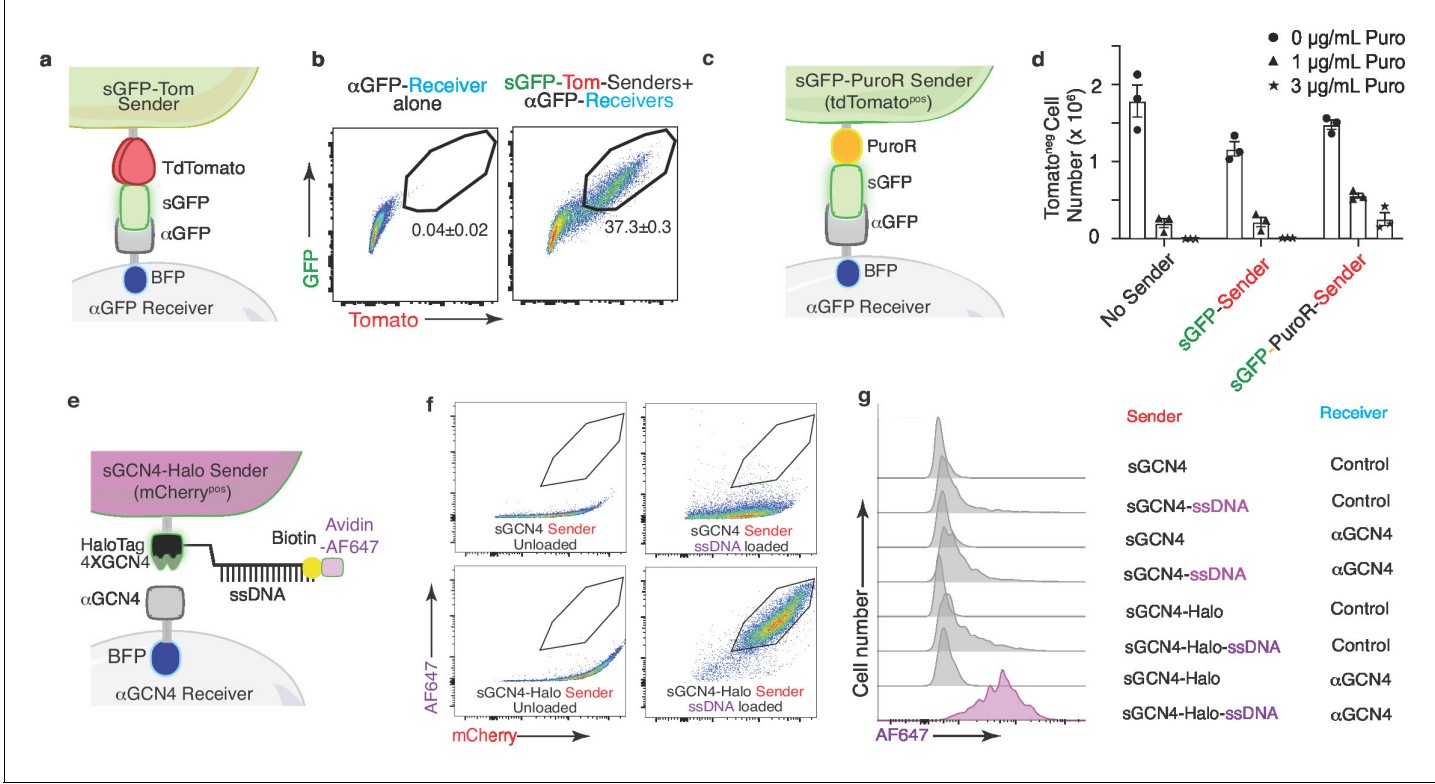

**Figure 8.** The G-baToN system can co-transfer cargo molecules to touched receiver cells. (**a**) Diagram of surface tdTomato-GFP (sGFP-Tom) co-transfer into touched receiver cells. Sender cells express sGFP-tdTomato and receiver cells express αGFP (tagged by intracellular BFP). (**b**) Representative FACS plots of αGFP 293 receiver cells cultured with the indicated sender cells at 1:1 ratio for 24 hr. Percent of GFP/Tomato dual-labeled receiver cells is indicated as mean +/- SD of BFP$^{pos}$ cells from triplicate cultures. (**c**) Diagram of GFP-PuroR co-transfer system. Sender cells express surface GFP associated with PuroR (marked by intracellular tdTomato), receiver cells express αGFP (tagged by intracellular BFP). PuroR: Gcn5-related N-acetyltransferase. (**d**) Co-transfer of GFP-PuroR from sGFP-PuroR sender cells to αGFP 293 receiver cells confers modest puromycin resistance to receiver cells. sGFP or sGFP-PuroR sender cells were co-cultured with αGFP 293 receiver cells for 24 hr at a 4:1 ratio before treatment with different doses of puromycin for 48 hr. tdTomato$^{neg}$PI$^{neg}$BFP$^{pos}$ cell numbers were counted via FACS. (**e**) Diagram of using GCN4-HaloTag sender to transfer ssDNA into αGCN4 receiver cells. Sender cells (marked by intracellular 2A-mCherry) express surface 4XGCN4 associated with a HaloTag, loaded with 5' HaloTag ligand, 3' biotin dual conjugated ssDNA (21 nt), then stained with Avidin-AF647. Receiver cells express αGCN4 (tagged by intracellular BFP). (**f**) Loading of sender cells with ssDNA. Representative FACS plots of 293 sender cells expressing either sGCN4 or sGCN4-Halo were loaded with 5' HaloTag-ligand, 3' biotin dual-conjugated ssDNA (21nt), then stained with Avidin-AF647. AF647 specifically labeled loaded sGCN4-Halo sender cells but not sGCN4 sender cells. (**g**) Representative plot of AF647 intensity in αGCN4 293 receiver cells co-cultured with GCN4-HaloTag 293 sender cells at a 1:1 ratio for 6 hr. Receiver cells were defined as mCherry$^{neg}$PI$^{neg}$BFP$^{pos}$ cells.

could be transferred to receiver cells. We generated sGFP-PuroR-expressing sender cells and found that co-culture of sGFP-PuroR sender cells with αGFP receiver cells led to moderate puromycin resistance of touched receiver cells (*Figure 8C,D*). Finally, loading of sGCN4-HaloTag sender cells with HaloTag-conjugated, AF647-coupled ssDNA prior to co-culture with αGCN4 receiver cells revealed successful co-transfer of fluorescently labeled ssDNA to receiver cells (*Figure 8E–G*). Thus, baToN systems enable contact-dependent transport of different macromolecules between cells.

## Discussion

Here, we developed and optimized a novel cell-cell interaction reporter system and showed that this G-baToN system can record diverse cell-cell interactions in a specific and sensitive manner. Our data document the ability of diverse primary cell types to serve as both sender and receiver cells, suggesting that the G-baToN system is not only simple, sensitive and rapid, but also generalizable. Multicolor derivatives of G-baToN enable qualitative and quantitative analyses of higher order interactions involving more than two cell types. Finally, the ability to co-transfer protein, DNA and chemical cargo suggests that this platform could be leveraged to manipulate target cell function.

Cancer cell-stromal cell interactions can be relatively stable (such as a cancer cell interacts with other cancer cells or stromal cells) or transient (such as cancer cell-immune cell interactions and circulating tumor cell (CTC) interactions with endothelial cells during metastasis). The G-baToN system labels receiver cells through transfer of cell surface GFP which, due to its lability, ensures only transient labeling. This is similar to other cell-cell interaction labeling systems (*Figure 1—figure supplement 2*). Transient labeling is sufficient to label stable cancer cell-stromal cell interactions and many other diverse cell-cell interactions when sender cells consistently express GFP (*Figure 1F*). This transient labeling should allow dynamic interactions to be detected, ensuring that the labeled receiver cells either are in contact with, or have recently interacted with sender cells.

Further optimization of the G-baToN systems could allow shorter or longer term labeling within different biological systems. For example, a G-baToN system where sGFP is inducible may allow physical interactions between cancer cells and stromal cells to be captured with even more precise temporal control. Conversely, we have shown that using chemical fluorophores significantly extends label persistence within receiver cells (*Figure 7E*), which can be used for longer term labeling of receiver cells that undergo dynamic interactions. Finally, due to the intrinsic attribute of G-baToN as a cell-cell contact dependent cargo transfer system (*Figure 8*), it should be possible to develop systems that allow for stable receiver cell labeling, perhaps through the transfer of site-specific recombinases (e.g. Cre or FLP) or programmable nucleases (e.g. Cas9) to genetically modify receiver cells upon contact. There will likely be some challenges when combing NLS-signal with sGFP. Nevertheless, additional G-baToN systems will greatly facilitate study of cell-cell interaction induced cell fate determination via contact-dependent lineage tracing.

As with other cell-cell interaction reporter systems, the G-baToN system relies on cell surface ligand-receptor recognition (*Supplementary files 1–2*). While LIPSTIC-based labeling is driven by endogenous ligand-receptor interactions, SynNotch and G-baToN systems rely on exogenous ligand-receptor pairs (*Morsut et al., 2016*; *Pasqual et al., 2018*). Consequently, these systems could stabilize normal cell-cell interactions. It is possible that tuning the affinity or expression level of the nanobody could minimize this effect. Inducible G-baToN systems may circumvent these issues, thus ensuring the recording of only bona fide interactions. Given that G-baToN-based cell-cell interaction reporter systems can be used as a discovery approach, the consequences of these interactions can be validated by orthogonal methods in the absence of exogenous ligand-receptor pairs. A recent paper used secreted cell-permeable mCherry for proximity labeling of cells. This system circumvents the problem of abnormal cell-cell interactions caused by ligand-receptor binding; however, this system labels all cells in proximity and lacks true cell-cell interaction labeling (*Ombrato et al., 2019*).

An interesting advantage of the G-baToN system is its ability to mediate cargo transfer. We demonstrated the feasibility of transferring small molecules (HaloTag ligand, *Figure 7*), functional proteins (puromycin resistant protein, *Figure 8C–D*), and non-protein macromolecules (ssDNA, *Figure 8E–G*). Transferred cargo proteins may be able to modify receiver cell signaling or promote cell death. In the future, additional design features could allow cancer cell-stromal cell interaction dependent drug delivery, cell-cell interaction facilitated sgRNA transfer between interacting cells, and digital recording of cell-cell interaction via DNA-barcode transfer. Thus, we expect the G-baToN system to facilitate an even wider array of discoveries about cell-cell interactions in cancer, across other physiological or pathological processes, and within different model organisms.

The simplicity of this two-component system, combined with its generalizability across cell types, excellent foreground to background ratio, and rapid labeling, should enable facile analysis of the dynamics of cellular interaction. These types of approaches have the potential to have a broad impact on our ability to understand the outputs of cell-cell interactions in cancer and various other biological systems.

## Materials and methods

### Cells, plasmids, and reagents

HEK-293T, B16-F10, A549, H460, and HUVEC cells were originally purchased from ATCC; HUASMC were purchased from PromoCell (C-12500); H82 cells were kindly provided by Julien Sage (Stanford School of Medicine); *KP* (238N1) and *KPT* (2985T2) lung adenocarcinoma cells were generated previously from tumors in $Kras^{LSL-G12D}$; $p53^{f/f}$ *and* $Kras^{LSL-G12D}$; $p53^{f/f}$; $R26^{LSL-Tom}$ mice. HEK-293T, 238N1,

2985T2 and B16-F10 cells were cultured in DMEM containing 10% FBS, 100 units/mL penicillin and 100 µg/mL streptomycin. A549, H460 and H82 cells were cultured in RPMI1640 media containing 10% FBS, 100 units/mL penicillin and 100 µg/mL streptomycin. HUVECs were cultured in Vascular Cell Basal Medium (ATCC, PCS-100–030) with Endothelial Cell Growth Kit (ATCC, PCS-100–041); HUASMC were cultured in Smooth Muscle Cell Growth Medium 2 (PromoCell, C-22062). All cell lines were confirmed to be mycoplasma negative (MycoAlert Detection Kit, Lonza).

Pitstop (ab120687) and Dyngo 4a (ab120689) were purchased from Abcam.

All plasmids used in this study are listed in *Supplementary file 1* and key plasmids for multiple G-baToN systems will be available on Addgene.

## Antibodies

Anti-GFP antibody was purchased from MyBioSource (MBS560494), anti-RFP antibody was purchased from Rockland (600-401-379), anti-human mitochondria antibody was purchased from Abcam (ab92824), anti-GAPDH antibody was purchased from Cell Signaling Technology (5174S).

## Lentiviral vector packaging

Lentiviral vectors were produced using polyethylenimine (PEI)-based transfection of 293 T cells with the plasmids indicated in *Supplementary file 1*, along with delta8.2 and VSV-G packaging plasmids in 150 mm cell culture plates. Sodium butyrate (Sigma Aldrich, B5887) was added 8 hr after transfection to achieve a final concentration of 20 mM. Medium was refreshed 24 hr after transfection. 20 mL of virus-containing supernatant was collected 36, 48, and 60 hr after transfection. The three collections were then pooled and concentrated by ultracentrifugation (25,000 rpm for 1.5 hr), resuspended overnight in 100 µL PBS, then frozen at −80°C.

## Generation of stable cell lines

Parental cells were seeded at 50% confluency in a six-well plate the day before transduction (day 0). The cell culture medium was replaced with 2 mL fresh medium containing 8 µg/mL hexadimethrine bromide (Sigma Aldrich, H9268-5G), 20 µL ViralPlus Transduction Enhancer (Applied Biological Materials Inc, G698) and 40 µL concentrated lentivirus and cultured overnight (Day 1). The medium was then replaced with complete medium and cultured for another 24 hr (Day 2). Cells were transferred into a 100 mm cell culture dish with appropriate amounts of puromycin (Dose used: 293T: 2 µg/mL; 238N1: 3 µg/mL; 2985T2: 2 µg/mL) and selected for 48 hr (Day 3). After selection, FACS analysis was performed using fluorescent markers indicated in *Supplementary file 2* for validation of selection efficiency.

## Transwell co-culture assay

The Corning Transwell polycarbonate membrane cell culture inserts were purchased from Corning Inc (3422: CS, Corning, NY). sGFP sender cells were seeded in the upper chamber inserts of the transwell ($1 \times 10^5$/insert). The inserts were then placed back into the plate pre-seeded with $1 \times 10^5$/well αGFP receiver cells and cultured in a humidified incubator at 37°C, with 5% $CO_2$ for 24 hr. sGFP sender and αGFP receiver cells co-cultured in the same plate under the same conditions were used as control. After 24 hr, the upper chamber inserts were removed, cells in the lower chamber were trypsinized and analyzed by flow cytometry.

## Live and fixed cell imaging

For live cell microscopy, $2 \times 10^4$ sGFP sender and $2 \times 10^4$ αGFP receiver cells were seeded into 35 mm FluoroDish Cell Culture Dishes (World Precision Instruments, FD35-100) and immediately imaged under a DeltaVision OMX (GE Healthcare) microscope with a 60x oil objective lens (Olympus) in a humidified chamber at 37°C with 5% $CO_2$. One image was taken per minute for 3 hr. Images were collected with a cooled back-thinned EM-CCD camera (Evolve; Photometrics).

For fixed cell microscopy, sender and receiver cells were seeded at the ratios indicated in *Supplementary file 2* with a total number of $1 \times 10^5$ cells onto Neuvitro-coated cover slips (Thermo Fisher Scientific, NC0301187) in a 12-well cell culture plate. 24 hr after co-culture, cells were fixed in 4% paraformaldehyde (PFA) PBS solution at room temperature for 10 min and washed with PBS and distilled water three times each, before mounting onto slides using 50% glycerol. Images were

captured using a Leica DMI6000B inverted microscope with an 40x oil objective lens. For quantification, GFP-containing receiver cells were counted. Multiple coverslips were analyzed across independent experiments (n = 10).

## Western blot

$5 \times 10^6$ sGFP sender and $5 \times 10^6$ αGFP receiver cells were co-cultured in a 100 mm cell culture dish for 24 hr. Cells were trypsinized, resuspended in FACS buffer containing propidium iodide (PI) (PBS, 2% FBS, 1 mM EDTA, and 1.5 μM PI). tdTomato$^{neg}$PI$^{neg}$ cells were sorted and lysed in RIPA buffer (50 mM Tris-HCl (pH 7.4), 150 mM NaCl, 1% Nonidet P-40, and 0.1% SDS) and incubated at 4°C with continuous rotation for 30 min, followed by centrifugation at 12,000 × rcf for 10 min. The supernatant was collected, and the protein concentration was determined by BCA assay (Thermo Fisher Scientific, 23250). Protein extracts (20–50 μg) were dissolved in 10% SDS-PAGE and transferred onto PVDF membranes. The membranes were blocked with 5% non-fat milk in TBS with 0.1% Tween 20 (TBST) at room temperature for 1 hr, followed by incubation with primary antibodies diluted in TBST (1:1000 for anti-GFP, anti-Tomato (RFP) and anti-human mitochondria (hu-Mito), 1:5000 for anti-GAPDH) at 4°C overnight. After three 10 min washes with TBST, the membranes were incubated with the appropriate secondary antibody conjugated to HRP diluted in TBST (1:10000) at room temperature for 1 hr. After three 10 min washes with TBST, Protein expression was quantified with enhanced chemiluminescence reagents (Fisher Scientific, PI80196).

## GFP/AF660 stability

To assess the stability of GFP and AF660 in touched receiver cells, $1 \times 10^7$ sGFP or sHalo-GFP sender cells were co-cultured with $1 \times 10^7$ αGFP receiver cells in a 150 mm cell culture dish for 6 hr. Cells were then trypsinized and resuspended in FACS buffer containing propidium iodide. mCherry$^{neg}$PI$^{neg}$BFP$^{pos}$ cells were sorted and $1 \times 10^5$ cells were re-plated in 12-well plate and cultured for 2, 4, 6, 8, 12, 24, and 48 hr in DMEM containing 10% FBS, 100 units/mL penicillin and 100 μg/mL streptomycin. GFP or AF660 intensity was assessed via FACS analysis of mCherry$^{neg}$PI$^{neg}$BFP$^{pos}$ cells and shown as Mean +/- SD of GFP/AF660 MFI in triplicate cultures.

## Puromycin-resistant protein transfer assay

To validate puromycin-resistant protein (GCN5-Related N-Acetyltransferases, PuroR) function in sender cells, $5 \times 10^6$ HEK-293T cells were transfected with sGFP-PuroR-PDGFR, sGFP-PDGFR or sGFP-PDGFR-IRES-PuroR in 100 mm cell culture dishes for 12 hr before re-plating into a 12-well plate ($1 \times 10^5$ cells/well). 24 hr after transfection, cells were treated with 1, 2, or 5 μg/mL puromycin for 24 hr. To count the number of viable receiver cells co-cultured with sGFP-PuroR sender cells, $2 \times 10^5$ αGFP receiver cells were co-cultured with $8 \times 10^5$ sGFP or sGFP-PuroR sender cells in a six-well plate. 24 hr after co-culture, cells were treated with 0, 1, and 3 μg/mL puromycin for 48 hr. Viable tdTomato$^{neg}$PI$^{neg}$BFP$^{pos}$ receiver cells were counted via FACS.

## Primary mouse cell isolation

Mouse (C57BL/6J, The Jackson Laboratory) lung, kidney, heart, hindlimb skeleton muscle, spleen, and liver tissue were dissected, cut into small pieces and digested in 5 mL tissue digest media (3.5 mL HBSS-Ca2+ free, 0.5 mL Trypsin-EDTA [0.25%], 5 mg Collagenase IV [Worthington], 25 U Dispase [Corning]) for 30 min in hybridization chamber at 37°C with rotation. Digestion is then neutralized by adding 5 mL ice cold Quench Solution (4.5 mL L15 media, 0.5 mL FBS, 94 μg DNase). Single-cell suspensions were generated by filtering through a 40 μM cell strainer, spinning down at 500 rcf for 5 min and washed with PBS twice.

For primary mouse lung epithelial cells, kidney epithelial cells and cardiomyocyte isolation and culture, the single-cell pellets were resuspended in 1 mL FACS buffer containing 1:300 dilution of anti-EpCam-AF467 (Biolegend, 118211) (for lung and kidney epithelial cell) or anti-Sirpa-AF467 (Biolegend, 144027) (for cardiomyocyte) antibody and incubated on ice for 20 min before FACS sorting. DAPI$^{neg}$EpCam$^{pos}$ or DAPI$^{neg}$Sirpa$^{pos}$ cells were sorted and seeded onto a 100 mm culture dish pre-coated with 5 μg/cm$^2$ Bovine Plasma Fibronectin (ScienCell, 8248).

For primary skeleton muscle cell, splenocyte and hepatocyte culture, the single cell pellets were resuspended in DMEM containing 20% FBS, 200 units/mL penicillin and 200 μg/mL streptomycin,

amphotericin and cultured in 100 mm culture dish at 37°C for 1 hr to remove fibroblast cells. The supernatant containing primary skeletal muscle cells, splenocytes and hepatocytes was then transferred into a new 100 mm culture dish precoated with 5 μg/cm$^2$ Bovine Plasma Fibronectin (ScienCell, 8248).

## Conjugation of HaloTag ligand to oligonucleotides

Oligonucleotides to be conjugated with the HaloTag ligand were synthesized with a 5' C12-linked amine and a 3' biotin group (IDT). Oligonucleotides were initially ethanol-precipitated and subsequently resuspended to 1 mM in conjugation buffer (100 mM Na2HPO4 (Sigma-Aldrich S9763), 150 mM NaCl (Thermo Fisher Scientific S271), pH 8.5). Resuspended oligos were combined with an equal volume of the HaloTag ligand succinimidyl ester (O4) (Promega P6751) resuspended in N,N- dimethylformamide (Sigma-Aldrich D4551) with a 30-fold molar excess of the ligand. Conjugation reactions were conducted overnight at room temperature with constant agitation prior to final cleanup via ethanol precipitation.

## Loading of HaloTag-expressing sender cells

Prior to loading with HaloTag-conjugated elements, sender cells were washed once in cold PBS following detachment and subsequently resuspended in cold Cell Staining Buffer (BioLegend 42021). For loading of HaloTag-conjugated fluorophores, sender cells were stained at a density of 1.00E+07 cells/mL on ice for 5 min in the presence of either 1 μM HaloTag-Alexa Fluor 488 (Promega G1001) or 3.5 μM HaloTag-Alexa Fluor 660 (Promega G8471). Stained sender cells were then washed twice in Cell Staining Buffer (500 rcf for 5 min at 4°C) prior to resuspension in growth media in preparation for co-culture.

For loading with HaloTag-conjugated oligonucleotides, sender cells were initially resuspended and incubated with 100 μg/mL salmon sperm DNA (Thermo Fisher Scientific 15632011). Sender cells were then incubated with 3.5 μM HaloTag-conjugated oligonucleotides (5AmMC12/TCTAGGCGCCCGGAATTAGAT/3Bio) and subsequently washed once. Oligonucleotide-loaded sender cells were then stained with 5 μg/mL streptavidin-conjugated Alexa Fluor 647 (Thermo Fisher Scientific S32357) for 30 min on ice. The loaded, stained sender cells were then washed twice and resuspended in growth media in preparation for co-culture.

## 3D-microfluidic cancer cell-endothelial cell co-culture

The master mold of microfluidic chips was fabricated using a 3D printer (Titan HD, Kudo3D Inc Dublin, CA). The surface of the molds was spray-coated with silicone mold release (CRC, cat. No.: 03300) and PDMS (poly-dimethyl siloxane, Sylgard 182, Dow Corning) was poured on it. After heat curing at 65°C for approximately 5 hr, the solidified PDMS replica was peeled off from the mold. Holes were made at both ends of each channel in the PDMS replica using a biopsy punch. The PDMS replica was then bonded to precleaned microscope glass slides (Fisher Scientific) through plasma treatment (Harrick Plasma PDC-32G, Ithaca, NY). Microfluidic chips were UV-treated overnight for sterilization before cell seeding.

A basement membrane extract (BME) hydrogel (Cultrex reduced growth factor basement membrane matrix type R1, Trevigen, Cat #: 3433–001 R1) was injected into the middle hydrogel channel of the chips placed on a cold pack and then transferred to rectangular 4-well cell culture plates (Thermo Scientific, Cat #: 267061) followed by incubation at 37°C in a cell culture incubator for 30 min for gelation. After gelation, 10 μL of human umbilical vein endothelial cells (HUVECs) resuspended at the density of ~1 × 10$^6$ cells/mL was injected to the blood channel of the chips and endothelial cell growth medium was added to the other side channel. After incubation for 3 hr for cells to adhere, medium in both side channels was replaced with fresh medium. The next day, samples were placed on a rocking see-saw shaker (OrganoFlow L, Mimetas) that generates a pulsatile bidirectional flow to mimic the dynamic native environment and cultured for 4 more days to form a complete endothelium. Cell culture medium was changed every other day. Then, medium in the blood channel of the chips was removed and 10 μL of sGFP-expressing lung adenocarcinoma cell at the density of ~1 × 10$^5$ cells/mL was injected and cultured for 24 hr before imaging. Images were captured using an EVOS fl auto imaging system (Life Technologies).

## Primary neuron and astrocyte cultures

Primary cortical neurons were dissociated from mouse (C57BL/6J, The Jackson Laboratory) E16.5 embryonic cortices into single-cell suspensions with a papain dissociation system (Worthington Biochemical Corporation). Tissue culture plates were coated with poly-L-lysine (0.1% w/v) before seeding cells. Neurons were grown in Neurobasal media (Gibco) supplemented with B-27 serum-free supplement (Gibco), GlutaMAX (Gibco), and penicillin-streptomycin (Gibco) in a humidified incubator at 37°C, with 5% $CO_2$. Half media changes were performed every 4–5 days. Primary astrocytes were dissociated from P0-P1 mouse cortices using the same papain dissociation methods as neurons, except the single-cell suspensions were then plated onto tissue culture plates without poly-L-lysine in DMEM with 10% FBS and penicillin-streptomycin. Primary astrocyte cultures were passaged using Accutase (Stemcell Technologies).

## Primary human T cell cultures

Blood from healthy donors collected in leukoreduction system (LRS) chambers was separated by Ficoll-Paque density gradient to obtain peripheral blood mononuclear cells (PBMCs). CD4[pos] and CD8[pos] T cells were isolated by negative selection using EasySep Human CD4+ T Cell Isolation Kit and EasySep Human CD8+ T Cell Isolation Kit (STEMCELL Technologies), respectively, according to the manufacturer's instructions. T cells were cultured for 3 days with CD3/CD28 Dynabeads (ThermoFisher Scientific) with 40 IU/mL IL-2 and spinoculated with lentivirus for 2 hr at 400 rcf in the presence of 8 µg/mL polybrene. T cells were expanded following transduction for two days in the presence of CD3/CD28 Dynabeads and 300 IU/mL IL-2 prior to use in assays. Transduced or untransduced CD4[pos] or CD8 [pos] T cells were co-cultured for 24 hr together with A549 cells. Following co-culture, cells were harvested, stained with antibodies against CD45, CD4, or CD8 (BioLegend) and analyzed on a LSRFortessa flow cytometer (BD Biosciences). Each condition was run in triplicate, and two independent experiments were conducted using T cells from different donors. For microscopy, A549 cells were co-cultured with CD4 [pos] T cells in glass-bottom plates (MatTek Corporation) and imaged on an LSM 700 confocal microscope (Zeiss).

## Statistical analysis

Sample or experiment sizes were estimated based on similar experiments previously performed in our laboratory, as well as in the literature. For the experiments in which two or more cell types are co-cultured, we used at least three samples per group for FACS analysis, at least 10 images were taken per group for image quantification. For the experiments in which cancer cells and T cells were transplanted into mice, we used at least three mice per group. In all the experiments reported in this study, no data points were excluded. No randomization was used in this study. There was no blinding method used to assign individuals to experimental groups.

Each experiment was repeated at least three times. All values are presented as mean ± SEM, with individual data points shown in the figure. Comparisons of parameters between two groups were made by two-tailed Student's t-tests. The differences among several groups was evaluated by one-way ANOVA with Tukey-Kramer post hoc evaluation. p-Values less than 0.05 and 0.01 were considered significant (*) or very significant (**), respectively.

## Acknowledgements

We thank the Stanford Shared FACS and Cell Sciences Imaging Facilities for technical support; A Orantes for administrative support, N Kipniss and YT Shue for critical reagents; D Feldser, J Sage, Y Chien, N Kipniss and members of the Winslow laboratory for helpful comments. We thank G Wahl, N Lytle, and L Li for ongoing discussions on future application of G-baToN systems. RT was supported by a Stanford University School of Medicine Dean's Postdoctoral Fellowship and a TRDRP Postdoctoral fellowship (27FT-0044). CWM was supported by the NSF Graduate Research Fellowship Program and an Anne T and Robert M Bass Stanford Graduate Fellowship. ILL was supported by a NIH F31 Predoctoral Fellowship (CA196029). WL was supported by a NCI Career Development Award (NIH K25-CA201545). This work was supported by NIH R01-CA175336 (to MMW), NIH R01-CA207133 (to MMW), NIH R01-CA230919 (to MMW), NIH R35-CA231997 and in part by the

Stanford Cancer Institute support grant (NIH P30-CA124435), S10OD012276 from the National Center for Research Resources (NCRR) and S10RR02743 from National Institutes of Health (NIH).

## Additional information

### Funding

| Funder | Grant reference number | Author |
|---|---|---|
| National Cancer Institute | R01-CA175336 | Monte M Winslow |
| National Cancer Institute | R01-CA207133 | Monte M Winslow |
| National Cancer Institute | R01-CA230919 | Monte M Winslow |
| Tobacco-Related Disease Research Program | 27FT-0044 | Rui Tang |
| National Science Foundation | Graduate Research Fellowship Program | Christopher W Murray |
| Stanford University | Anne T. and Robert M. Bass Fellowship | Christopher W Murray |
| National Institutes of Health | CA196029 | Ian L Linde |
| National Cancer Institute | K25-CA201545 | Wonjae Lee |
| Stanford University School of Medicine | Dean's Postdoctoral Fellowship | Rui Tang |

The funders had no role in study design, data collection and interpretation, or the decision to submit the work for publication.

### Author contributions

Rui Tang, Conceptualization, Data curation, Formal analysis, Supervision, Funding acquisition, Validation, Investigation, Visualization, Methodology, Writing - original draft, Project administration, Writing - review and editing; Christopher W Murray, Ian L Linde, Data curation, Writing - review and editing; Nicholas J Kramer, Zhonglin Lyu, Min K Tsai, Leo C Chen, Hongchen Cai, Aaron D Gitler, Edgar Engleman, Wonjae Lee, Data curation; Monte M Winslow, Conceptualization, Supervision, Funding acquisition, Writing - original draft, Writing - review and editing

### Author ORCIDs

Rui Tang (iD) https://orcid.org/0000-0002-6950-9580
Christopher W Murray (iD) https://orcid.org/0000-0002-6551-2053
Nicholas J Kramer (iD) http://orcid.org/0000-0003-4557-8343
Leo C Chen (iD) http://orcid.org/0000-0002-4950-0757
Aaron D Gitler (iD) http://orcid.org/0000-0001-8603-1526
Edgar Engleman (iD) http://orcid.org/0000-0002-2096-9279
Monte M Winslow (iD) https://orcid.org/0000-0002-5730-9573

### Ethics

Animal experimentation: This study was performed in strict accordance with the recommendations in the Guide for the Care and Use of Laboratory Animals of the National Institutes of Health. All of the animals were handled according to approved institutional animal care and use committee ( the Administrative Panel on Laboratory Animal Care (APLAC)) protocols (26696) of Stanford University. The protocol was approved by the Committee on the Ethics of Animal Experiments of Stanford University (Permit Number: A3213-01). Every effort was made to minimize suffering.

### Decision letter and Author response

Decision letter https://doi.org/10.7554/eLife.61080.sa1
Author response https://doi.org/10.7554/eLife.61080.sa2

# Additional files

## Supplementary files

• Supplementary file 1. Plasmids used in each experiment in this study. For each figure panel we list how the cells are referred to in the figure (Cell Name in Figure), the parental plasmid, the plasmid used to generate that cell line (N-terminal extracellular domain-transmembrane domain-cytoplastic domain; N-TM-C), whether they are the same plasmid used in a previous figure (Repeated Cell Line), whether the cells were selected with puromycin, and whether the nanobody had a Myc tag. The cells used in each experiment are listed in *Supplementary file 2*.

• Supplementary file 2. Cells used in each experiment in this study. For each figure panel we list how the cells are referred to in the figure (Cell Name in Figure), the parent cell line or cell type, whether cDNA expression was stable (via lentiviral transduction) or transient (via transfection), the transgene expressed in those cells, whether they are the same cell line used in a previous figure (Repeated Cell Line), whether they are senders or receivers, whether the expression of GFP, BFP, mCherry or tdTomato is cell surface or intracellular, and whether the nanobody has a Myc tag. For each experiment, we list the Sender to Receiver cell ratio and the co-culture time. The vectors used for cDNA expression in each cell type in each experiment are listed in *Supplementary file 1*.

• Transparent reporting form

## Data availability

All data generated or analysed during this study are included in the manuscript and supporting files.

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
