## [Decision Letter]

**Acceptance summary:**

This manuscript describes a reporter system for studying cell-cell interactions. Multiple specific cell-cell interactions are examined to provide proof of principle for reporter use, including multi-color systems for reporting on complex interactions between 3 or more cell types. One could envision this system being used for a wide variety of downstream applications, in both biological and bioengineering studies.

**Decision letter after peer review:**

Thank you for submitting your article "G-baToN: a versatile reporter system for cancer cell-stromal cell interactions" for consideration by *eLife*. Your article has been reviewed by Richard White as the Senior Editor, a Reviewing Editor, and three reviewers. The following individuals involved in review of your submission have agreed to reveal their identity: Reuben Shaw (Reviewer #1); Francisco Sanchez-Rivera (Reviewer #2); Kıvanç Birsoy (Reviewer #3).

The reviewers have discussed the reviews with one another and the Reviewing Editor has drafted this decision to help you prepare a revised submission.

Summary:

This manuscript describes a reporter system for studying cell-cell interactions. Multiple specific cell-cell interactions are examined to provide proof of principle for reporter use, including multi-color systems for reporting on complex interactions between 3 or more cell types. One could envision this system being used for a wide variety of downstream applications, in both biological and bioengineering studies.

Revisions:

The reviewers make some experimental suggestions, which could be included at your discretion, but are not required. Please revise the manuscript at a minimum to address the points listed next. Full reviewer comments are then provided for your information only and do not require a full response as you prepare your revised manuscript.

1. Please provide any additional detail on the 1-6 hour timepoints in Figure 1f. Is there a burst of transfer around 2 hour or any specific average of time after first contact? (see comment from Reviewer 1).

2. Please comment on whether it might be possible to transfer the Cre-recombinase to a different cell population to turn on a floxed reporter in a neighboring cell. Along these lines, also please comment on the point from Reviewer 2 regarding only a minor population of cells being active as receivers. Can you comment whether this is because the system inefficient, or short lived, or the transferred proteins are not stable?

3. Comment on single copy (low MOI) versus multiple copy (high MOI) lentiviral transduction used for the system as Reviewer 2 points out this is relevant for future applications. Also comment on leakiness of the system, and issues are half-life of fluorophores to highlight additional potential limitations of this system.

4. Consider mentioning in the abstract that the system could be used for studying neuron-astrocytes interactions to broaden impact.

5. Please include plasmid sequences in the supplementary material.

6. Reviewer 3 commented the resistance to puromycin in Figure 8 was weak, please comment on this in the revised manuscript.

Reviewer #1:

In this interesting technology development manuscript from the Winslow lab, the authors have created a very adaptable and flexible reporter system for studying cell-cell interactions. The authors have extensively controlled for each feature of the reporter and the data is quantitative and compelling throughout. Multiple specific kinds of cell-cell interactions are examined in this tour-de-force of proof of principle examples. Cancer cell – neuron, cancer cell – T cells, cancer cell- endothelial cells, and even neurons – astrocytes are examined here. Multi-color systems for reporting on complex interactions between 3 or more cell types are examined, which speaks to the more likely functionality one would observe in vivo. Beyond the initial nanobody design, the authors end the study with HaloTag baton systems and very interesting data reporting the co-transfer of cargo molecules into target cells, including Tomato, PuroR, and ssDNA. One could envision this system being used for a wide variety of downstream applications, in a wide variety of biological and bioengineering fields. It is that last point which makes this manuscript a compelling one to publish in *eLife*, as it will be of broad interest to several fields.

Reviewer #2:

I) Relevance

Cell-cell interactions play critical roles across all of biology, including during normal organismal development and disease initiation, progression, and resolution. Achieving a comprehensive mechanistic understanding of how specific cell-cell interactions are established and disentangling the biological processes that are regulated by such cell-cell interactions ultimately requires that one is able to (1) track and quantitatively measure the initiation of specific cell-cell interactions; (2) follow the fate of interacting cells over time; (3) isolate and characterize interacting cells; (4) probe interacting cells; and (5) ideally be able to do all of the above in a spatiotemporal manner. Progress in this area has been limited by the lack of faithful experimental tools that allow for rapid, specific, and ideally long-lived labeling of cell-cell interactions, particularly in the in vivo setting. Recent advances in the areas of synthetic biology (e.g. synNotch receptors; PMID: 26830878 and 27693353) and chemical biology (e.g. Sortase-based LIPSTIC labeling approaches; PMID: 29342141) have opened the door for mechanistically probing cell-cell interactions in a reasonably specific manner. More recently, a system that employed secreted fluorescent proteins (sLP-mCherry; PMID: 31462798) was developed to label neighboring groups of cells in specific niches, some of which were shown to interact directly. All of these systems have drawbacks, including the requirement for multiple complementary components expressed in each of the interacting cells and potential unspecific labeling of artificial cell-cell interactions due to inappropriate expression or activity or some of the modular components that form these systems.

II) General concept and approach

In this manuscript, Tang et al., present G-baToN, a novel cell-cell interaction platform that relies on direct cell to cell transfer of a fluorescent protein by a 'Sender' cell, which leads to rapid (but temporary) labeling of a 'Receiver' cell. G-baToN is largely based on the synNotch concept, whereby a 'Sender' cell expresses a fluorescent reporter (Surface GFP, or sGFP) in the surface and a 'Receiver' cell expresses a synthetic Notch-based nanobody that recognizes sGFP, which in turn leads to activation of the Notch pathway, intracellular cleavage and release of a transcription factor, and induction of a synthetic gene reporter. However, instead of relying on a synthetic gene reporter in 'Receiver' cells, G-baToN exploits the concept of ligand internalization, whereby a nanobody-sGFP complex is internalized into cells, leading to sGFP-based labeling of the 'Receiving' cell. Thus, G-baToN does not need an additional reporter component in 'Receiver' cells; it only requires that they express the sGFP nanobody in their surface. Furthermore, the sGFP nanobody does not need to be engineered as a synNotch receptor (i.e. it does not need to be fused to a cleavable Notch-based transcription factor). Therefore, G-baToN is a simpler and more versatile system relative to the original synNotch system and also to the Sortase-based LIPSTIC system (which relies on specific receptor-ligand pairs and in exogenous substrates for cell-cell interaction labeling).

III) Novelty and comparison to other similar methods

As mentioned above, G-baToN is largely based on the synNotch concept, relying on sGFP and anti-sGFP nanobody expression in the surface of 'Sender' and 'Receiver' cells, respectively. Therefore, G-baToN is not novel from that point of view, even though it employs carefully optimized engineered versions of these components (see Figure 2). Where G-baToN really shines is in its simplicity relative to both synNotch and LIPSTIC systems. Although G-baToN still relies on expression of specific synthetic components in the surface of 'Sender' and 'Receiver' cells, it dramatically simplifies the identification and isolation of 'Receiver' cells since (1) no reporter gene needs to be turned on in 'Receiver' cells; (2) no exogenous substrate needs to be provided in order for cell labeling to take place; and (3) due to being a 2-component system, it is less prone to 'system breakdown' (e.g. synNotch systems can easily break via spontaneous mutations in the internal cleavage sites, in the transcription factor that turns on the reporter gene in 'Receiver' cells, or in the promoter or coding sequence of the reporter gene itself). Furthermore, due to its simplicity and modularity, G-baToN can be seamlessly expanded to allow for isolation of both 'Sender' and 'Receiver' interacting cells by essentially endowing 'Receiver' cells with 'Sender' capability and vice versa (see Figure 6). Specifically, one can simply engineer 'Sender' cells to express (1) a specific antibody in the surface targeting a surface fluorescent protein encoded in the 'Receiver' cell and (2) a different fluorescent protein expressed in its own cell surface. On the other hand, 'Receiver' cells can be engineered to express (1) a surface antibody targeting the 'Sender' surface fluorescent protein and (2) a fluorescent protein expressed in its cell surface recognized by the 'Sender' surface antibody (see. Figure 6). Furthermore, the authors show that one can append synthetic tags and fluorophores to the fluorescent proteins, which can increase both the half-life and the dynamic range of cell-cell interaction labeling, and that the inherent modularity and combinatorial nature of the system allows for further detection and isolation of higher-order cell-cell interactions (e.g. 'Receiver' cells that have interacted with 2 independent 'Sender' cells) (see Figure 7). Collectively, this allows for efficient bidirectional reporting and isolation of both simple and higher-order cell-cell interactions, which will undoubtedly open the door for more comprehensive molecular and cellular characterization of each cell within an interaction unit. Lastly, the authors show that the modular nature of G-baToN can allow for interaction-dependent transfer of various types of molecular cargoes (antibiotic resistance cassettes, ssDNA molecules), albeit with an apparent variable efficiency. Thus, G-baToN is indeed a much, much more versatile system when compared to synNotch and LIPSTIC.

In a very similar vein, a recent study in Nature described a novel system based on local secretion of fluorescent proteins (in this case a secreted lipid permeable mCherry, called sLP-mCherry; PMID: 31462798) that also allows for labeling of cell-cell interactions, albeit with lower specificity than that achieved by synNotch, LIPSTIC, and G-baToN. To my surprise, there was no mention of this study, which should definitely be referenced and embedded within the discussion and benchmarking of G-baToN (see Figure 1—figure supplement 2), as it is of direct relevance to this and future approaches, particularly in the context of in vivo cancer studies. Furthermore, this would allow readers to think more creatively about how to potentially integrate these complementary systems to further advance the concepts behind mapping cell-cell interactions.

IV) Specific comments

I think this is overall a very good study that warrants publication as a Tools and Resources Article in *eLife*. Even though it builds upon existing concepts and approaches (e.g. synNotch), it does so in an intelligent and systematic manner, leading to a significantly optimized and highly versatile platform that can be deployed across a variety of scenarios to detect, isolate, and characterize cell-cell interactions, as well as leveraged for perturbing interacting cells via cell-to-cell cargo transfer. Due to its versatility and modularity, the system could easily be combined with inducible-reversible technologies (e.g. Tet-inducible promoters, Cre/loxP, Flp/Frt, degrons, etc.) to allow for exquisite spatiotemporal control of cell-cell interaction labeling, isolation, and perturbation.

Reviewer #3:

In this manuscript, the authors describe a new reporter system that exploits a cell surface antigen and a nanobody interaction enabling directional labeling between diverse cell types. The authors present sufficient evidence for the utility of the technique (different cell types, different cargos, wide range of fluorophores). This is a rigorous study that deserves to be published in *eLife*. While there have been previous attempts to provide solutions for studying cell-cell interaction (The authors mentioned some of these tools), I believe this will be a useful tool as well. Time will tell the practical aspect of this and previous methods.

- One aspect that is lacking in the paper is whether cell-cell interaction through their method identifies functional differences between cells that interact vs others that do not. It is not really clear whether cells being close to each other, if labeled, is sufficient to cause any functional change in both cells. Are these interactions productive or are they just randomly choosing cells proximal to each other? I understand that the manuscript does not focus on a particular phenotype, however, even a simple RNA-seq of sorted populations would answer this question.

- Given the half-life of the fluorophores, one should expect a decrease in labeling over time, which may lead to missing many interacting cells. It would be a great if the authors discuss potential limitations of using cargo as a way to label cells.

---

## [Author Response]

Revisions:The reviewers make some experimental suggestions, which could be included at your discretion, but are not required. Please revise the manuscript at a minimum to address the points listed next. Full reviewer comments are then provided for your information only and do not require a full response as you prepare your revised manuscript.1. Please provide any additional detail on the 1-6 hour timepoints in Figure 1f. Is there a burst of transfer around 2 hour or any specific average of time after first contact? (see comment from Reviewer 1).

We thank the reviewer for noting the rapid GFP transfer at early time points. To better resolve the dynamics of GFP transfer at 0-6 hours, we collected αGFP-receiver (HEK293T, at 1:1 ratio) at 15 min, 30 min, 1 hour, 2 hours, 4 hours, 6 hours, 8 hours, and 12 hours of co-culture with sGFP-senders (KPT). As shown in new Figure 1—figure supplement 1g, there is indeed a large amount of transfer around 2 hours. However, we think that this is most likely driven by technical aspects of the experiment. Senders and receivers are trypsinized, counted, and mixed for co-culture. It normally takes 1-2 hours for cells to attach to culture plate, where cell-cell interaction can be stabilized. Thus, we believe that transfer may occur even more rapidly upon cell-cell contact. However, we do note that transfer kinetics could be cell type-dependent.

2. Please comment on whether it might be possible to transfer the Cre-recombinase to a different cell population to turn on a floxed reporter in a neighboring cell.

As we now discuss in our manuscript (Discussion), due to the capacity of G-baToN to mediate the transfer of cargo (Figure 8), we believe that it should be possible to co-transfer Cre recombinase (or other site specific or programable nucleases). However, there will likely be some challenges when combing NLS-Cre with sGFP. The NLS could hinder sGFP expression or transfer efficiency. Furthermore, for transferred Cre to get to the nucleus it will have to detach from the sGFP-TM anchor following transfer. There are multiple systems that could solve the problems (e.g. transfer a CreERT2). Thus, co-transfer a bioactive Cre into a floxed reporter receiver cell with high efficiency remains an important future direction. In fact, in collaboration with Geoffrey Wahl's lab at the Salk Institute, we are developing modified versions of G-baToN enabling indelible labeling of receiver cells using site-specific recombinase technology.

Along these lines, also please comment on the point from Reviewer 2 regarding only a minor population of cells being active as receivers. Can you comment whether this is because the system inefficient, or short lived, or the transferred proteins are not stable?

We do not think that this stems from variable degrees of transfer efficiency of the G-baToN system, but rather due to differences in copy number, transgene expression, and αGFP presentation in different cell types. Although we have not directly assessed the relationship between αGFP expression and GFP transfer efficiency, we have noted that αGFP expression generally correlates with GFP transfer. Even though, among all the cell types we tested, as long as the αGFP expression level can be detected by FACS (either through surface Myc tag or intracellular BFP), the cell can be used as receiver. We included a variety of different cell types as receivers, including human cell lines (HEK293T, most figures), endothelial cells and smooth muscle cells (Figure 3), T cells

(Figure 4), neurons (Figure 5), lung and kidney epithelial cells, skeleton muscle cells, astrocytes, splenocytes (Figure 5-figure supplement 2), and human cancer cells (Figure 5—figure supplement 2).

3. Comment on single copy (low MOI) versus multiple copy (high MOI) lentiviral transduction used for the system as Reviewer 2 points out this is relevant for future applications. Also comment on leakiness of the system, and issues are half-life of fluorophores to highlight additional potential limitations of this system.

We thank the reviewers for bringing up this practical concern regarding the G-baToN technology. For transductions, we used high MOI (>1) to generate all stable cell lines (sender and receiver). Beyond copy number, transgene expression level can be controlled in multiple ways (e.g. promotor activity, codon optimization) which are likely more important for driving sGFP and nanobody expression. Furthermore, surface presentation of sGFP or αGFP may be enhanced via optimization of signal peptides in different cell types. G-baToN used exogenous fluorophores to label cells, which gives both advantage in increasing signal-to-noise ratio (minimal leakiness) and disadvantage in a shorten half-life for labeling.

For leakiness, because the G-baToN system is fluorescence-transfer-based, it is more a foreground to background (natural autofluorescence) consideration rather leakiness.

For half-life of fluorophore, protein fluorophore will have a shorter half-life (half-life around 6 hours), but we show evidence that a chemical fluorophore persists longer. Researchers can optimize the G-baToN system to fit their needs:

To record transient interaction: we expect to (1) adopt an inducible expression system to limit the labeling of interacting cells during a defined time period; (2) transfer recombinase to achieve permeant labeling.

To record stable interaction: we have shown that as long as sender cell presents, GFP signal can sustain for a long time (Figure 1f).

4. Consider mentioning in the abstract that the system could be used for studying neuron-astrocytes interactions to broaden impact.

Thank you for this suggestion. We have now changed the title into “A versatile system to record cell-cell interactions” and noted the application of G-baToN in the context of neuron-astrocyte interactions in the Abstract.

5. Please include plasmid sequences in the supplementary material.

There is a large number of plasmids used in this study. Rather than directly providing all of the sequences, we have compiled publicly accessible Benchling links for these plasmids in supplementary file 1. We have noted this in the Methods section of the manuscript. We are happy to also include plasmids sequences as supplemental material if the journal prefers. We will deposit all of the key plasmids at Addgene.

6. Reviewer 3 commented the resistance to puromycin in Figure 8 was weak, please comment on this in the revised manuscript.

We thank the reviewer for pointing this out. We acknowledge in the text that the resistance conferred by transferred PuroR is not dramatic. This is likely influenced by (1) impaired transfer efficiency caused by extra cargo proteins; and (2) the short half-life for transferred PuroR in receiver cells. However, as a proof-of-principle experiment, this result still provides evidence that G-baToN can transfer cargo protein with biological

function. Further efforts will be required to optimize cargo protein transfer efficiency and maintain activity in receiver cells.

Reviewer #1:In this interesting technology development manuscript from the Winslow lab, the authors have created a very adaptable and flexible reporter system for studying cell-cell interactions. The authors have extensively controlled for each feature of the reporter and the data is quantitative and compelling throughout. Multiple specific kinds of cell-cell interactions are examined in this tour-de-force of proof of principle examples. Cancer cell – neuron, cancer cell – T cells, cancer cell- endothelial cells, and even neurons – astrocytes are examined here. Multi-color systems for reporting on complex interactions between 3 or more cell types are examined, which speaks to the more likely functionality one would observe in vivo. Beyond the initial nanobody design, the authors end the study with HaloTag baton systems and very interesting data reporting the co-transfer of cargo molecules into target cells, including Tomato, PuroR, and ssDNA. One could envision this system being used for a wide variety of downstream applications, in a wide variety of biological and bioengineering fields. It is that last point which makes this manuscript a compelling one to publish in eLife, as it will be of broad interest to several fields.Reviewer #2:I) RelevanceCell-cell interactions play critical roles across all of biology, including during normal organismal development and disease initiation, progression, and resolution. Achieving a comprehensive mechanistic understanding of how specific cell-cell interactions are established and disentangling the biological processes that are regulated by such cell-cell interactions ultimately requires that one is able to (1) track and quantitatively measure the initiation of specific cell-cell interactions; (2) follow the fate of interacting cells over time; (3) isolate and characterize interacting cells; (4) probe interacting cells; and (5) ideally be able to do all of the above in a spatiotemporal manner. Progress in this area has been limited by the lack of faithful experimental tools that allow for rapid, specific, and ideally long-lived labeling of cell-cell interactions, particularly in the in vivo setting. Recent advances in the areas of synthetic biology (e.g. synNotch receptors; PMID: 26830878 and 27693353) and chemical biology (e.g. Sortase-based LIPSTIC labeling approaches; PMID: 29342141) have opened the door for mechanistically probing cell-cell interactions in a reasonably specific manner. More recently, a system that employed secreted fluorescent proteins (sLP-mCherry; PMID: 31462798) was developed to label neighboring groups of cells in specific niches, some of which were shown to interact directly. All of these systems have drawbacks, including the requirement for multiple complementary components expressed in each of the interacting cells and potential unspecific labeling of artificial cell-cell interactions due to inappropriate expression or activity or some of the modular components that form these systems.II) General concept and approachIn this manuscript, Tang et al., present G-baToN, a novel cell-cell interaction platform that relies on direct cell to cell transfer of a fluorescent protein by a 'Sender' cell, which leads to rapid (but temporary) labeling of a 'Receiver' cell. G-baToN is largely based on the synNotch concept, whereby a 'Sender' cell expresses a fluorescent reporter (Surface GFP, or sGFP) in the surface and a 'Receiver' cell expresses a synthetic Notch-based nanobody that recognizes sGFP, which in turn leads to activation of the Notch pathway, intracellular cleavage and release of a transcription factor, and induction of a synthetic gene reporter. However, instead of relying on a synthetic gene reporter in 'Receiver' cells, G-baToN exploits the concept of ligand internalization, whereby a nanobody-sGFP complex is internalized into cells, leading to sGFP-based labeling of the 'Receiving' cell. Thus, G-baToN does not need an additional reporter component in 'Receiver' cells; it only requires that they express the sGFP nanobody in their surface. Furthermore, the sGFP nanobody does not need to be engineered as a synNotch receptor (i.e. it does not need to be fused to a cleavable Notch-based transcription factor). Therefore, G-baToN is a simpler and more versatile system relative to the original synNotch system and also to the Sortase-based LIPSTIC system (which relies on specific receptor-ligand pairs and in exogenous substrates for cell-cell interaction labeling).III) Novelty and comparison to other similar methodsAs mentioned above, G-baToN is largely based on the synNotch concept, relying on sGFP and anti-sGFP nanobody expression in the surface of 'Sender' and 'Receiver' cells, respectively. Therefore, G-baToN is not novel from that point of view, even though it employs carefully optimized engineered versions of these components (see Figure 2). Where G-baToN really shines is in its simplicity relative to both synNotch and LIPSTIC systems. Although G-baToN still relies on expression of specific synthetic components in the surface of 'Sender' and 'Receiver' cells, it dramatically simplifies the identification and isolation of 'Receiver' cells since (1) no reporter gene needs to be turned on in 'Receiver' cells; (2) no exogenous substrate needs to be provided in order for cell labeling to take place; and (3) due to being a 2-component system, it is less prone to 'system breakdown' (e.g. synNotch systems can easily break via spontaneous mutations in the internal cleavage sites, in the transcription factor that turns on the reporter gene in 'Receiver' cells, or in the promoter or coding sequence of the reporter gene itself). Furthermore, due to its simplicity and modularity, G-baToN can be seamlessly expanded to allow for isolation of both 'Sender' and 'Receiver' interacting cells by essentially endowing 'Receiver' cells with 'Sender' capability and vice versa (see Figure 6). Specifically, one can simply engineer 'Sender' cells to express (1) a specific antibody in the surface targeting a surface fluorescent protein encoded in the 'Receiver' cell and (2) a different fluorescent protein expressed in its own cell surface. On the other hand, 'Receiver' cells can be engineered to express (1) a surface antibody targeting the 'Sender' surface fluorescent protein and (2) a fluorescent protein expressed in its cell surface recognized by the 'Sender' surface antibody (see. Figure 6). Furthermore, the authors show that one can append synthetic tags and fluorophores to the fluorescent proteins, which can increase both the half-life and the dynamic range of cell-cell interaction labeling, and that the inherent modularity and combinatorial nature of the system allows for further detection and isolation of higher-order cell-cell interactions (e.g. 'Receiver' cells that have interacted with 2 independent 'Sender' cells) (see Figure 7). Collectively, this allows for efficient bidirectional reporting and isolation of both simple and higher-order cell-cell interactions, which will undoubtedly open the door for more comprehensive molecular and cellular characterization of each cell within an interaction unit. Lastly, the authors show that the modular nature of G-baToN can allow for interaction-dependent transfer of various types of molecular cargoes (antibiotic resistance cassettes, ssDNA molecules), albeit with an apparent variable efficiency. Thus, G-baToN is indeed a much, much more versatile system when compared to synNotch and LIPSTIC.In a very similar vein, a recent study in Nature described a novel system based on local secretion of fluorescent proteins (in this case a secreted lipid permeable mCherry, called sLP-mCherry; PMID: 31462798) that also allows for labeling of cell-cell interactions, albeit with lower specificity than that achieved by synNotch, LIPSTIC, and G-baToN. To my surprise, there was no mention of this study, which should definitely be referenced and embedded within the discussion and benchmarking of G-baToN (see Figure 1—figure supplement 2), as it is of direct relevance to this and future approaches, particularly in the context of in vivo cancer studies. Furthermore, this would allow readers to think more creatively about how to potentially integrate these complementary systems to further advance the concepts behind mapping cell-cell interactions.

Thanks for the suggestion. The sLP-mCherry system is not contact-depend and lacks specificity, thus we decided not to include it in Figure 1—figure supplement 2. However, we add some discussion about it (Discussion section).

IV) Specific commentsI think this is overall a very good study that warrants publication as a Tools and Resources Article in eLife. Even though it builds upon existing concepts and approaches (e.g. synNotch), it does so in an intelligent and systematic manner, leading to a significantly optimized and highly versatile platform that can be deployed across a variety of scenarios to detect, isolate, and characterize cell-cell interactions, as well as leveraged for perturbing interacting cells via cell-to-cell cargo transfer. Due to its versatility and modularity, the system could easily be combined with inducible-reversible technologies (e.g. Tet-inducible promoters, Cre/loxP, Flp/Frt, degrons, etc.) to allow for exquisite spatiotemporal control of cell-cell interaction labeling, isolation, and perturbation.Reviewer #3:In this manuscript, the authors describe a new reporter system that exploits a cell surface antigen and a nanobody interaction enabling directional labeling between diverse cell types. The authors present sufficient evidence for the utility of the technique (different cell types, different cargos, wide range of fluorophores). This is a rigorous study that deserves to be published in eLife. While there have been previous attempts to provide solutions for studying cell-cell interaction (The authors mentioned some of these tools), I believe this will be a useful tool as well. Time will tell the practical aspect of this and previous methods.- One aspect that is lacking in the paper is whether cell-cell interaction through their method identifies functional differences between cells that interact vs others that do not. It is not really clear whether cells being close to each other, if labeled, is sufficient to cause any functional change in both cells. Are these interactions productive or are they just randomly choosing cells proximal to each other? I understand that the manuscript does not focus on a particular phenotype, however, even a simple RNA-seq of sorted populations would answer this question.

We acknowledge that our current manuscript focuses on the technical development of this system, without providing new biological insights. We agree that this will be important in subsequent studies.

- Given the half-life of the fluorophores, one should expect a decrease in labeling over time, which may lead to missing many interacting cells. It would be a great if the authors discuss potential limitations of using cargo as a way to label cells.

We agree that the short half-life of GFP is a double-edged sword, it is good because labeling is transient and it is also bad because labelling is transient.